# Spatial structure of disordered proteins dictates conductance and selectivity in nuclear pore complex mimics

**Adithya N Ananth[1†], Ankur Mishra[2†], Steffen Frey[3], Arvind Dwarkasing[1], Roderick Versloot[1], Erik van der Giessen[2], Dirk Görlich[3]\*, Patrick Onck[2]\*, Cees Dekker[1]\***

[1]Department of Bionanoscience, Kavli Institute of Nanoscience, Delft University of Technology, Delft, Netherlands; [2]Zernike Institute for Advanced Materials, University of Groningen, Groningen, Netherlands; [3]Max Planck Institute for Biophysical Chemistry, Göttingen, Germany

**Abstract** Nuclear pore complexes (NPCs) lined with intrinsically disordered FG-domains act as selective gatekeepers for molecular transport between the nucleus and the cytoplasm in eukaryotic cells. The underlying physical mechanism of the intriguing selectivity is still under debate. Here, we probe the transport of ions and transport receptors through biomimetic NPCs consisting of Nsp1 domains attached to the inner surface of solid-state nanopores. We examine both wildtype FG-domains and hydrophilic SG-mutants. FG-nanopores showed a clear selectivity as transport receptors can translocate across the pore whereas other proteins cannot. SG mutant pores lack such selectivity. To unravel this striking difference, we present coarse-grained molecular dynamics simulations that reveal that FG-pores exhibit a high-density, nonuniform protein distribution, in contrast to a uniform and significantly less-dense protein distribution in the SG-mutant. We conclude that the sequence-dependent density distribution of disordered proteins inside the NPC plays a key role for its conductivity and selective permeability.
DOI: https://doi.org/10.7554/eLife.31510.001

**\*For correspondence:**
dgoerli@gwdg.de (DGö);
p.r.onck@rug.nl (PO);
C.Dekker@tudelft.nl (CD)

[†]These authors contributed equally to this work

**Competing interests:** The authors declare that no competing interests exist.

## Introduction

The nuclear envelope (NE) separates the nucleus of eukaryotic cells from the cytosol. NE-embedded nuclear pore complexes (NPCs) allow for the exchange of molecules such as RNA, metabolites, and proteins between the two compartments. NPCs are giant structures with a molecular mass of around 100 MDa, composed of about 30 different types of proteins named nucleoporins (Nups) (*Hurt and Beck, 2015*; *Hoelz et al., 2016*; *Schwartz, 2016*). NPCs are equipped with a barrier that is permeable for molecules of up to 30 kDa or ~5 nm in diameter, but blocks the passage of larger ones (*Popken et al., 2015*; *Schmidt and Görlich, 2016*; *Timney et al., 2016*). Shuttling nuclear transport receptors (NTRs) can overcome this size-limit and traverse the NPC, carrying along cargoes with diameters of up to 40 nm (*Panté and Kann, 2002*; *Lowe et al., 2010*), thus endowing the pore with a selective permeability barrier. Nups that contain phenylalanine-glycine (FG) repeats (FG-Nups) (*Hurt, 1988*) are crucial for this remarkable selectivity, suggesting that the NTR transport is mediated by hydrophobic interactions. The FG-repeat domains are intrinsically disordered, bind NTRs during facilitated translocation (*Iovine et al., 1995*; *Bayliss et al., 1999*), and form the NPC permeability barrier (*Frey and Görlich, 2007*; *Patel et al., 2007*). The question of how FG domains create a permeability barrier and at the same time greatly favor the passage of NTRs is one of the central questions in molecular cell biology.

Many different models have been proposed to explain the selective transport of NTRs through NPCs, including the virtual-gate model (*Rout et al., 2000*; *Rout et al., 2003*), the reversible-collapse (or polymer-brush) model (*Lim et al., 2007*), the reduction-of-dimensionality (*Peters, 2005*) and 'molecular velcro' (*Schleicher et al., 2014*) model, the hydrogel model (*Ribbeck and Görlich, 2001*; *Frey and Görlich, 2007*), the Kap-centric model (*Lim et al., 2015*; *Kapinos et al., 2017*), and the forest model (*Yamada et al., 2010*). However, no consensus has been reached on one prevailing model.

A typical NPC comprises about 10–12 different FG Nups (in copy numbers of 8 to 32), yielding about 5000 FG motifs per NPC. One of the most abundant and best-studied FG Nups is *S. cerevisiae* Nsp1 (*Hurt, 1988*). Here, we specifically address the importance of FG domains for NPCs by comparing the transport properties of Nsp1-coated biomimetic NPCs with analogs that employ an Nsp1 mutant in which the hydrophobic amino acids F, I, L and V are replaced by hydrophilic serines (S), thus creating an 'SG' Nsp1 variant. To realize this, we employ an approach that combines biophysics experiments and coarse-grained molecular dynamics (MD) simulations. For the experiments, we utilize the approach of biomimetic NPCs (*Caspi et al., 2008*; *Jovanovic-Talisman et al., 2009*) based on the solid-state-nanopore platform (*Kowalczyk et al., 2011b*). Solid-state nanopores, basically small holes in a silicon nitride membrane, are single-molecule sensors based on ion-current readout. As a robust, modular, and label-free technique (*Dekker, 2007*), nanopores provide a powerful platform to study NPCs in a bottom-up approach. Using these nanopore-based biomimetic NPCs, we here investigate the ion transport through such pores at various diameters as well as compare the selectivity of NPCs with Nsp1-FG domains with those made of the Nsp1-SG mutant.

Important insight in the nanoscopic structure of these biomimetic NPCs is obtained by complementing the in-vitro measurements with in-silico simulation results of an experimentally-calibrated one-bead-per-amino-acid MD model (*Ghavami et al., 2013*; *Ghavami et al., 2014*). The key feature of the model is that it is fine enough to represent the amino-acid sequence of each Nsp1-FG domain and its SG-mutant, but coarse enough to capture the collective behavior of all FG-domains inside the biomimetic nanopore (that contains over 80,000 amino-acids altogether). The model is used to establish the nonhomogeneous density distribution inside the pores of different diameters and to shed light on the relation between ion conductance and FG-domain density. Furthermore, using umbrella sampling, the energy barrier of inert cargos and transport receptors is calculated to address the difference in selectivity and permeability between nanopores lined with Nsp1 and its mutant. The in-vitro and in-silico data agree very well and highlight the role of hydrophobic interactions in nuclear transport. Our findings identify how the sequence-dependent spatial structure of the disordered FG domains affects the conductance and establishes the NPC's selective permeability.

## Results

### Conductance of Nup-coated biomimetic NPCs

To study the structural and transport properties of FG domains within biomimetic NPCs, we used self-assembled-monolayer chemistry to graft the domains to the surface of the solid-state nanopore, using a C-terminal cysteine for surface attachment. A scheme of the attachment chemistry is shown in *Figure 1—figure supplement 1*. To build the minimal NPC mimic, we first examined the important and well-studied FG domain (*Hurt, 1988*) from S. cerevisiae: Nsp1[1-601] (65.7 kDa) (*Figure 1A*), which has a highly cohesive N-terminus and a charged non-cohesive C-terminal part (*Ader et al., 2010*; *Yamada et al., 2010*). Additionally, we studied an Nsp1 mutant, in which the hydrophobic amino acids F, I, L, V have been replaced by the hydrophilic amino acid serine (S). Given the abundance of F compared to I, L and V, the major change in sequence is the replacement of the FG and FxFG motifs into SG and SxSG motifs, thus converting the Nsp1 FG-domain into a Nsp1 SG-domain (see Materials and methods for the exact amino-acid sequence of the wildtype and mutant Nsp1). In earlier studies, it was shown that the mutated Nsp1-SG domain was unable to form a hydrogel-like structure (*Frey et al., 2006*; *Patel et al., 2007*; *Ader et al., 2010*).

Here, we study how this affects the conductance of the biomimetic NPCs as well as their selective properties. Once the nanopore was coated with the Nsp1-FG domains (further called Nsp1 in short) and Nsp1-SG domains (further called Nsp1-S), current (I) versus voltage (V) curves for each pore were recorded at physiological salt conditions and applied voltages from −200 mV to 200 mV. All

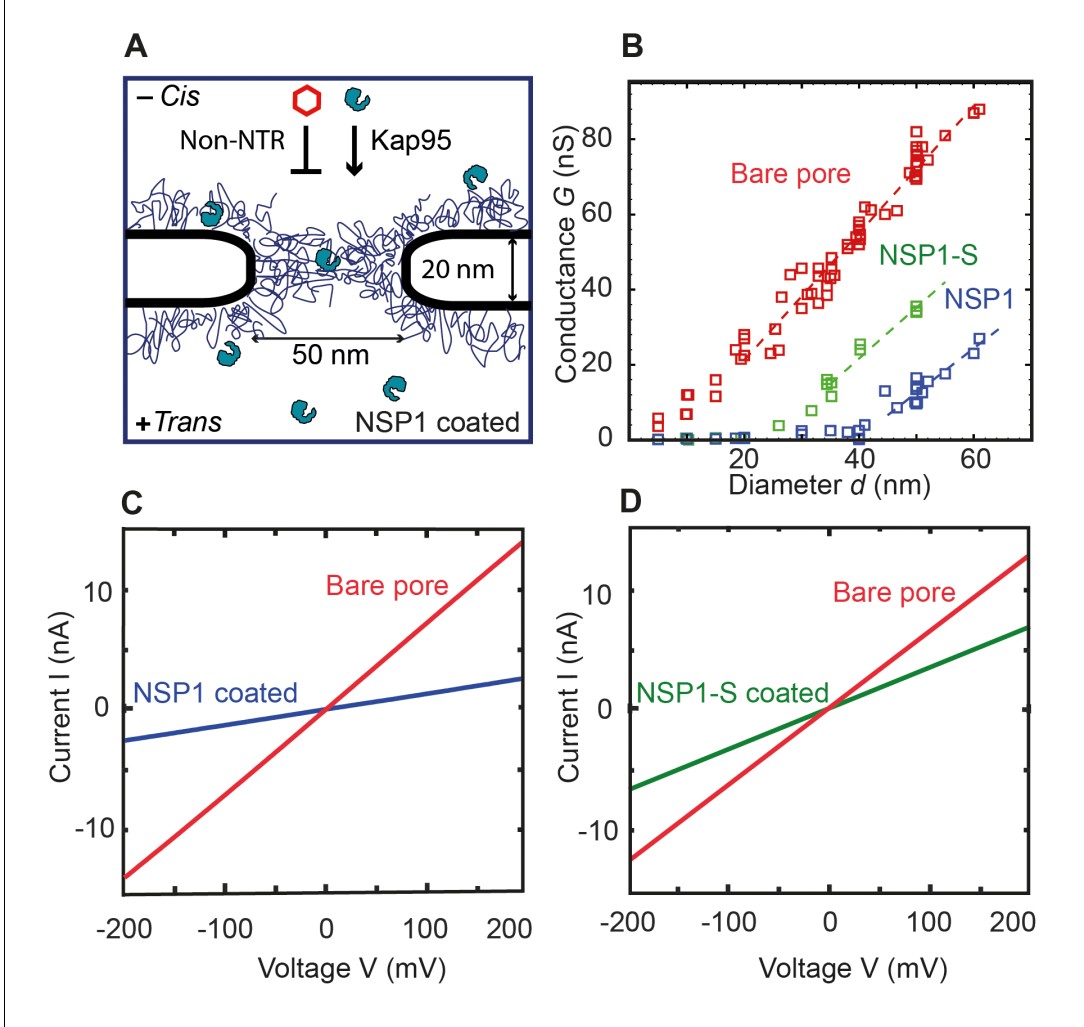

**Figure 1.** Coating a nanopore with FG-Nups reduces the pore conductivity. (**A**) Schematic of the biomimetic NPC where yeast FG-Nup Nsp1 is coated onto a solid-state nanopore of diameter 50 nm and thickness of 20 nm. Kap95, a yeast importer, can pass through the barrier, whereas most other proteins such as tCherry fail to pass through the pores. (**B**) Conductance versus pore diameter for bare pores (red), Nsp1-coated pores (blue), and Nsp1-S-coated pores (green). The conductance is low (<4 nS) for small-diameter biomimetic pores, below a threshold diameter 41 ± 2 nm and 26 ± 3 nm, for Nsp1 and Nsp1-S respectively. Above this threshold diameter, the conductance increases linearly with slope similar to that of the bare pore conductance. Dashed lines are linear guides to the eye. (**C and D**) Current vs voltage curves for a 50 nm pore before (red) and after Nsp1 coating (blue). The conductance drops by about 80% after coating, confirming a high density of Nsp1 inside the nanopore. (**D**) Current vs voltage curves for a mutant Nsp1-S-coated (green) 50 nm pore. Here the conductance drops by about 50% compared to the bare pore (red).

DOI: https://doi.org/10.7554/eLife.31510.002

The following figure supplements are available for figure 1:

**Figure supplement 1.** Self-assembled monolayer surface chemistry for covalently attaching Nsp1 and Nsp1-S to Silicon Nitride membrane.

DOI: https://doi.org/10.7554/eLife.31510.003

**Figure supplement 2.** Current power spectral density of a bare pore and nanopores coated with Nsp1 and Nsp1-S, versus frequency.

DOI: https://doi.org/10.7554/eLife.31510.004

**Figure supplement 3.** Transmission electron microscopy (TEM) images of bare and Nsp1-coated pores.

DOI: https://doi.org/10.7554/eLife.31510.005

**Figure supplement 4.** Histogram of the conductance of individual Nsp1 molecules translocating through a bare pore, and the associated scatter plot of conductance versus translocation time.

DOI: https://doi.org/10.7554/eLife.31510.006

**Figure supplement 5.** Histogram of the conductance of individual Nsp1-S molecules translocating through a bare pore, and the associated scatter plot of conductance versus translocation time.

DOI: https://doi.org/10.7554/eLife.31510.007

*Figure 1 continued on next page*

Figure 1 continued

**Figure supplement 6.** Example QCM-D traces for the surface functionalization of Nsp1 (**A**) and Nsp1-S (**B**) on silicon nitride coated with APTES and Sulfo-SMCC.

DOI: https://doi.org/10.7554/eLife.31510.008

**Figure supplement 7.** Average protein densities of a Nsp1-S coated solid-state nanopore of diameter 45 nm with grafting distance 5.7 nm (green) and 5.9 nm (red).

DOI: https://doi.org/10.7554/eLife.31510.009

pores showed a linear IV response, see *Figure 1C,D* for examples. The IV characteristics of both the Nsp1 and Nsp1-S grafted pores are linear but with a lower slope than for the bare pores, indicating, as expected, a reduced ion conductance due to the presence of the Nups. The attachment of Nups to the nanopore also increased the low-frequency 1/*f* noise compared to bare pores (See *Figure 1— figure supplement 2*). Transmission electron micrographs of Nsp1-coated pores further supported the presence of Nups within the nanopores (*Figure 1—figure supplement 3*). The linearity of the IV curves indicates that the Nsp1 and Nsp1-S coat was not significantly affected by the applied voltage. For the Nsp1-coated pores, the conductance $G = I/V$ dropped about 80% after coating Nsp1 (*Figure 1C*). For pores coated with Nsp1-S, the current drop was lower, about 50% when compared with bare pores (*Figure 1D*). The difference in the current blockade points towards a different volumetric arrangement of the proteins inside the nanopore, thus emphasizing the difference in the amino acid sequence of Nsp1 and Nsp1-S.

Biomimetic NPCs have the advantage that, unlike natural NPCs, the pore diameter can be varied as a free parameter. We compared the ionic conductance $G = I/V$ of bare pores with Nsp1 and Nsp1-S coated pores for various pore diameters $d$ (*Figure 1B*). For bare pores, a conductance of $G = 6–88$ nS was measured for pore diameters ranging from 5 to 60 nm. We observed a slightly non-linear increase of conductance at small pore sizes, followed by a near-linear relation for wide pores. This is in accordance with the well-established non-linear $G(d)$ relation for cylindrical SiN pores (*Hall, 1975*; *Kowalczyk et al., 2011a*):

$$G(d) = \sigma_{\mathrm{bare}}\left[4l/\left(\pi d^2\right) + 1/d\right]^{-1}, \tag{1}$$

where the first term in the denominator accounts for the pore resistance and the second for the access resistance (the latter being dominant at large pore diameters). Here, $l = 20$ nm is the height of the pore and $\sigma_{\mathrm{bare}}$ is the conductivity of the ions through the bare pore, which was fitted to be equal to $2.2 \pm 0.2$ nS/nm (average ±standard deviation), in close agreement with the experimental value of $2.3 \pm 0.3$ nS/nm from bulk conductivity measurements.

For Nsp1-coated pores, the conductance data show a radically different behaviour, with two rather distinct regimes of ion conductivity above and below an apparent threshold diameter of $d_{\mathrm{Nsp1}} = 41 \pm 2$ nm. The current measured for pores with a diameter ranging from 5 nm to 41 nm showed a very low conductance of $G = 0.2$ to 4 nS (see *Figure 3—figure supplement 1*). Nsp1-coated pores with a diameter larger than 41 nm conduct ions with a much larger conductance. These observations are consistent with previously published results for biomimetic NPC's with human FG domains (*Kowalczyk et al., 2011b*). When we coat the pores with the Nsp1-S mutant, we observed a qualitatively similar non-linear $G(d)$ behaviour as for the Nsp1-coated pores, but with a much lower threshold diameter $d_{\mathrm{Nsp1-S}} = 23 \pm 3$ nm.

## Molecular dynamics calculations of the FG domain density distribution

In order to gain a microscopic understanding of the FG domain structures that underlie these nonlinear in vitro conductance data, we developed a coarse-grained MD model of the biomimetic nanopores with embedded FG domains. The MD model of the domains is based on a one-bead-per-amino-acid representation that distinguishes between all 20 amino acids (see *Figure 2A*) (*Ghavami et al., 2014*). The model takes into account hydrophobic and electrostatic interactions between the amino acids, as well as the screening effect of free ions and the polarity of the solvent. The model has been shown to accurately predict (within 20% error) the Stokes radii of a wide range of FG domains and FG domain segments (*Ghavami et al., 2014*), including the low-charge Nsp1[1-172] and high-charge Nsp1[173-603] FG segments (*Yamada et al., 2010*). Nanopores were modeled as cylinders of height 20 nm (see Materials and methods) constructed from inert beads of 3 nm diameter as depicted in *Figure 2B*. The Nsp1 and Nsp1-S

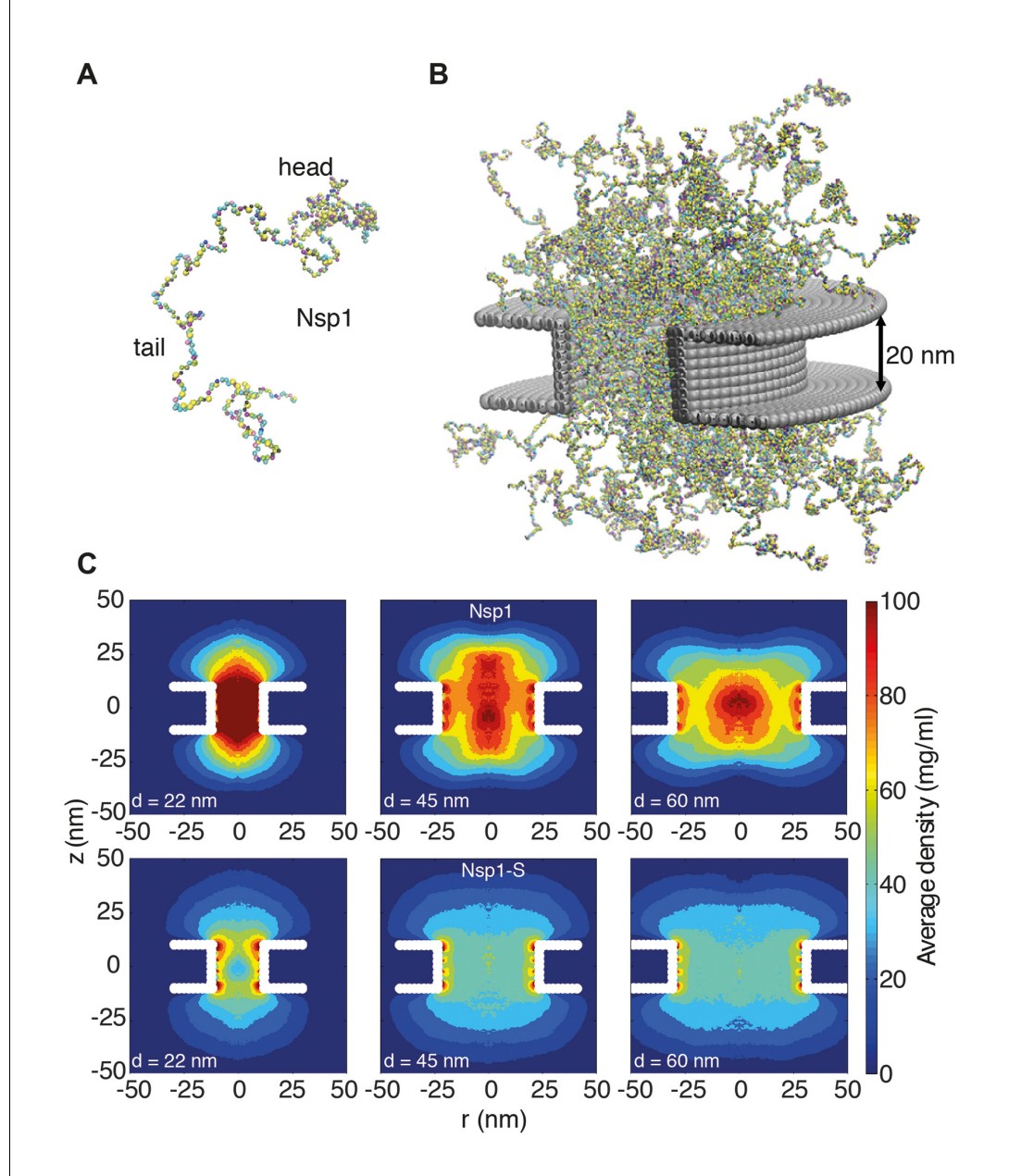

**Figure 2.** Coarse-grained molecular dynamics results of Nup density distributions in Nsp1 and Nsp1-S pores of varying diameter. (**A**) Coarse-grained one-bead-per-amino-acid representation of Nsp1; the different colors of the beads represent the 20 different amino acids. The collapsed-coil N-terminal 'head' region is visible at the top right. (**B**) Multiple Nsp1s tethered inside a cylindrical pore of height 20 nm and a diameter of 45 nm with anchor points spaced according to a fully triangulated (close-packed) distribution with a spacing of 5.7 nm. (**C**) Time-averaged *r-z* density distribution of Nsp1-coated nanopores (top row) with diameters 22 nm, 45 nm and 60 nm; and similarly for Nsp1-S (bottom row). These data show denser structures for the smaller pores and much lower densities for Nsp1-S compared to the wildtype Nsp1. The nups are coated on the inner surface of the cylindrical nanopores at a close-packed triangular spacing of 5.7 nm.

DOI: https://doi.org/10.7554/eLife.31510.010

The following figure supplements are available for figure 2:

**Figure supplement 1.** Top: The configuration of Nsp1 consists of a 'collapsed coil' head (in blue) and a 'extended coil' stalk region (multi-colored; each color represents a different amino acid).

DOI: https://doi.org/10.7554/eLife.31510.011

**Figure supplement 2.** Simulation results for the density distribution in Nup98-coated biomimetic pores.

DOI: https://doi.org/10.7554/eLife.31510.012

were anchored in a close-packed triangular lattice with an average grafting density of 1 per 28 nm$^2$, corresponding to an average grafting distance of 5.7 nm. This grafting distance was experimentally estimated using two independent techniques (see Materials and methods, *Figure 1—figure supplement 4*; *Figure 1—figure supplement 5*; *Figure 1—figure supplement 6*), and further confirmed in experiments on denatured proteins in guanidinium HCl (Materials and methods). The 1 per 28 nm$^2$ grafting density matches well with the surface area per FG Nup in a yeast NPC of about 24 to 32 nm$^2$ and is close to the density that was reported for Nsp1 assembled in vitro on a planar surface (*Eisele et al., 2013*).

We thus computed the time-averaged amino acid mass density distribution of the nanopores that were coated with Nsp1 or Nsp1-S, for pore diameters ranging from 22 to 60 nm. *Figure 2C* shows the axisymmetric ($r$, $z$) density distribution in the pores, averaged in the circumferential direction. The mass density inside the central cylindrical region of the larger Nsp1 pores is much higher (70–100 mg/ml), than that for the mutant (50 mg/ml), as can also be seen in the $z$-averaged ($-10$ nm $<z$ $< 10$ nm) radial density distribution in *Figure 3A*. Interestingly, we observed that the Nsp1 pores clearly feature a maximum density at the central axis ($r = 0$, see *Figure 3*), which is possibly related to the high percentage of hydrophobic residues, relative to charged residues, in the head group of the wildtype Nsp1. The Nsp1-S data show a striking difference in density distribution: much more uniform and less dense, which is likely to be caused by the lower number of hydrophobic residues compared to the wildtype Nsp1 (see *Figure 2C* and *Figure 3A*).

To further explore the partitioning of Nsp1 in the pore based on amino-acid sequence, we study the localization of its head and tail groups inside the nanopore. Nsp1 has a collapsed coil N-terminal segment that is hydrophobic, low in charge and rich in FG-repeats, forming a small cohesive 'head'. The C-terminus domain – which is bound to the nanopore surface – has a high charge-to-hydrophobicity ratio and has a repulsive, extended coil ('stalk') conformation (*Yamada et al., 2010*) (see *Figure 2—figure supplement 1*). Our results show that for Nsp1 the heads are rather localized, forming a cohesive structure around the central pore axis for the 45 (see *Video 1*) and 60 nm pores. In contrast, the Nsp1-S heads show a much more widespread distribution (see *Video 2*), reflecting their higher charge-to-hydrophobicity ratio.

The Stokes radii ($R_S$) of the head and stalk region are 3.2 nm and 6.5 nm, respectively, as computed by us (*Ghavami et al., 2014*) and measured by Yamada and coworkers (*Yamada et al., 2010*). To investigate the role of these two segments in establishing the high-density structure, we plot the density distribution of only those amino-acids that are part of the N-terminal head region in *Figure 2—figure supplement 1*. For the mutant we see a more wide-spread distribution, whereas for Nsp1 the heads are rather localized, forming a cohesive structure around the central pore axis for the largest pore sizes (for the smaller pore size of 22 nm the geometric confinement is so large that the Nsp1 heads are pushed out from the core of the pore). It is interesting to note that the radii of gyration of the head and stalk region in isolation are similar as when they are part of one Nsp1 molecule. However, the radius of gyration of Nsp1 is less than the sum of the radius of gyration of the head and tail, indicating that the head and tail do interact but retain their individual conformation (see *Video 3*), even when tethered together (see SI *Video 1*). This also carries over to their conformation inside the pore, albeit with one difference: the radius of gyration of the stalk region is enlarged by 30%, whereas the radius of gyration of the head again remains unchanged. This is most likely caused by the stronger lateral constraints of the stalks at the anchor points (C-terminal), while the N-terminal heads have more freedom.

In terms of amino acid sequence and pore partitioning it is interesting to compare these Nsp1 pores also with nanopores lined with the Nup98 FG domain (498 amino acids), studied before (*Kowalczyk et al., 2011b*). The Nup98 FG domain has a low charge-to-hydrophobicity ratio, resulting in a collapsed structure, and it is grafted on the pore surface at a density of 1 per 49 nm$^2$ (*Kowalczyk et al., 2011b*). The 2D ($r$,$z$) and radial density distribution, depicted in *Figure 2—figure supplement 2*, show a profoundly different behaviour: the Nup98 FG pore shows a very dense (300 mg/ml) ring-like structure that forms already at relatively small pore sizes (25–30 nm), while the protein density vanishes towards the pore centre. In contrast, Nsp1 and its mutant form a pore-filling Nup network that is retained up to pore diameters larger than 60 nm. The key observation is that, consistent with experiments (*Kowalczyk et al., 2011b*), the ionic conductance through the Nup98 pores only commences when a central conduit has opened up in the nonconductive high-density ring structure, which contrast Nsp1 and Nsp1-S pores that are filled by a uniform protein network of relative low density that supports ion flow throughout (see below).

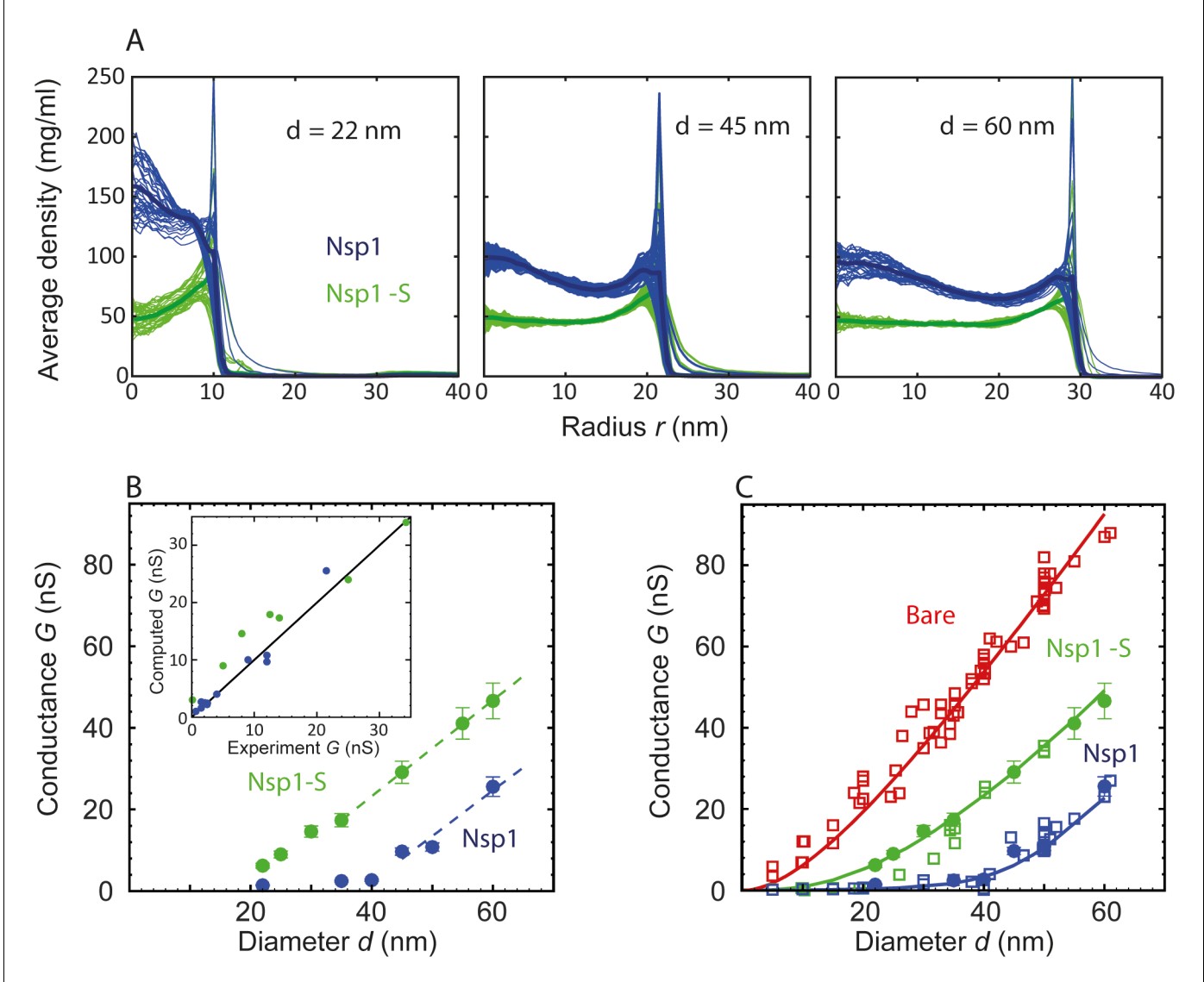

**Figure 3.** Radial density distribution and conductance data for Nsp1 and Nsp1-S biomimetic pores. (**A**) Radial protein density distribution for biomimetic nuclear pores with pore diameters of 22 nm, 45 nm, and 60 nm, for pores coated with Nsp1 (blue) and Nsp1-S (green). All data are taken within the height of the cylinder (20 nm; $-10$ nm $< z < 10$ nm) that is divided into 20 equally spaced discs of thickness 1 nm each. Each of the 20 curves represented in each panel shows the radial density distribution for that specific z location. (**B**) Modeling results for the conductance as a function of pore diameter for Nsp1-coated pores (blue) and Nsp1-S-coated pores (green). The dashed lines are linear guides to the eye. The inset shows a comparison between the computed and the experimental conductance. (**C**) Conductance versus pore diameter for the experimental (open symbols) and modeling data (closed symbols). For Nup-coated pores, the conductance is low ($G < 4$ nS) for small diameters, but it increases strongly with a non-linear dependence on pore diameter beyond ~40 nm for Nsp1 and beyond ~20 nm for Nsp1-S. At larger diameters the conductance increases almost linearly with a slope slightly smaller than that of the bare pore, with G-values of tens of nS. The red solid line corresponds to *Equation (1)* for the bare pore and the green and blue solid lines correspond to *Equation (2)* with the conductivities for the access and pore regions obtained by fitting the numerical results using sigmoidal functions (see *Figure 3—figure supplement 2* ).

DOI: https://doi.org/10.7554/eLife.31510.013

The following figure supplements are available for figure 3:

**Figure supplement 1.** Conductance as a function of pore diameter below 40 nm.
DOI: https://doi.org/10.7554/eLife.31510.014

**Figure supplement 2.** The conductivity in the pore region ($\sigma_{\text{pore}}$) and access region ($\sigma_{\text{access}}$) for the simulated nanopores lined with Nsp1 (blue circles; panel **A**) and Nsp1-S (green circles; panel **B**) plotted as a function of pore diameter.
DOI: https://doi.org/10.7554/eLife.31510.015

**Figure supplement 3.** The conductivity and conductance for the simulated nanopores lined with Nup98 as a function of pore diameter.

*Figure 3 continued on next page*

*Figure 3 continued*

DOI: https://doi.org/10.7554/eLife.31510.016

**Figure supplement 4.** Computed conductance vs the experimentally measured conductance, for (**A**) Nsp1 (blue circles) and Nsp1-S (green circles), and (**B**) Nup98 (black circles).

DOI: https://doi.org/10.7554/eLife.31510.017

## A density-based conductance relation for biomimetic NPCs

Based on the computational results of the inhomogeneous mass distributions of the biomimetic NPCs, we now calculate the modified ion conductance through the pores. For nanopores coated with FG domains, the presence of the proteins hinders electrical transport, thus reducing the effective conductivity of the medium in the pore. To account for this, we introduce $\sigma_{\mathrm{access}}$ and $\sigma_{\mathrm{pore}}$ to describe the ion conductivity of the access region and the pore, respectively. The total conductance of the nanopore can then be written as:

$$G(d) = \left[4l/\left(\pi d^2 \sigma_{\mathrm{pore}}\right) + 1/(d\sigma_{\mathrm{access}})\right]^{-1}. \tag{2}$$

To calculate the effective conductivity $\sigma_{\mathrm{pore}}$ for a specific pore diameter, we make use of the radial density distributions $\rho(r)$ of the Nups inside the pore, that is, averaged over the range $-10$ nm $< z < 10$ nm (*Figure 3A*). The ion conductivity is taken equal to the bare-pore ion conductivity $\sigma_{\mathrm{bare}}$ = 2.2 nS/nm for regions where the Nup density is zero, and assumed to decrease in proportion to the local protein density, $\sigma(r) = \sigma_{\mathrm{bare}}(1 - \rho(r)/\rho_{crit})$, where $\rho_{crit}$ is a free parameter found to be equal to 85 mg/ml from a fit to the data. The conductivity is taken to be zero at and beyond that critical density. Then by radially integrating $\sigma(r)$, we obtain the conductivity of the pore as

$$\sigma_{\mathrm{pore}} = \left(4/\pi d^2\right) \int\limits_{r=0}^{r=\frac{d}{2}} 2\pi r\sigma(r)dr. \tag{3}$$

A related expression is used to similarly calculate the access conductivity ($\sigma_{\mathrm{access}}$). Hyun and coworkers probed the size of the access region and showed that for nanopores with a similar $l/d$ ratio (1.5) as used here, the access resistance is only affected in a region closer than 40 nm from the center of the pore (*Hyun et al., 2012*). For smaller values of $l/d$, the size of the access region was found to decrease. Therefore, we define the access region to extend from 10 nm $< |z| <$ 40 nm, and we use this range to calculated the density distribution as a function of $r$. We performed a sensitivity analysis for the size of the access region and observed that by decreasing the size with a factor as large as 3, the maximal change in all computed conductance values was only found to be 13%, showing that the results are not sensitive to changes in the size of the access region. Finally, we use these pore ($\sigma_{\mathrm{pore}}$) and access conductivities ($\sigma_{\mathrm{access}}$) to calculate the total pore conductance described by the modified conductance relation (*Equation 2*) for the different diameters ranging from 22 to 60 nm, resulting in *Figure 3B*.

*Figure 3B* shows a dependence of $G$ on $d$ that is strikingly similar to that of the experimental data (cf. *Figure 1B*), featuring two distinct regimes of ion conductance, at low and high

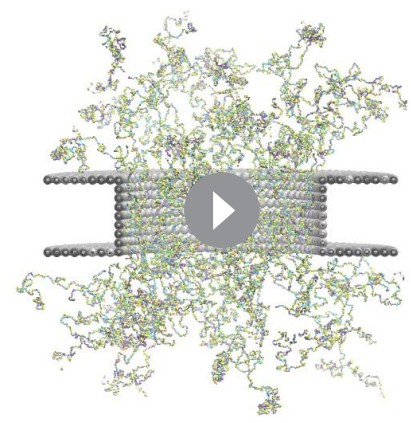

**Video 1.** A short trajectory of a coarse-grained one-bead-per-amino-acid molecular dynamics simulation of a biomimetic nanopore with a diameter of 45 nm coated with Nsp1, corresponding to *Figure 2C* (top row) of the main manuscript. Only half the simulation box is shown for better visibility. The movie is prepared with the Visual Molecular Dynamics (VMD) software.

DOI: https://doi.org/10.7554/eLife.31510.018

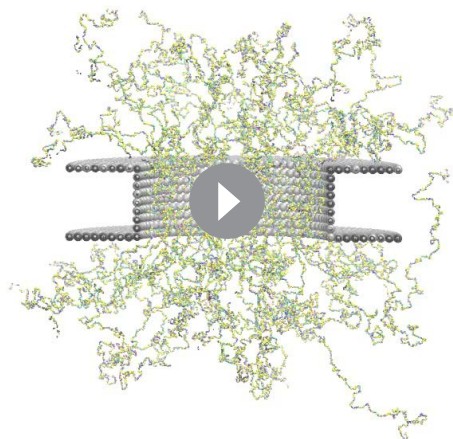

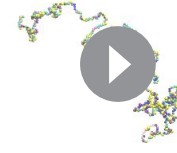

**Video 2.** A short trajectory of a coarse-grained one-bead-per-amino-acid molecular dynamics simulation of a biomimetic nanopore with a diameter of 45 nm coated with the mutant Nsp1-S, corresponding to *Figure 2C* (bottom row) of the main manuscript. Only half the simulation box is shown for better visibility The movie is prepared with the Visual Molecular Dynamics (VMD) software.
DOI: https://doi.org/10.7554/eLife.31510.019

**Video 3.** A short trajectory of a coarse-grained one-bead-per-amino-acid molecular dynamics simulation of an isolated Nsp1. The cohesive head group at the N-terminus is located at the right. The movie is prepared with the Visual Molecular Dynamics (VMD) software.
DOI: https://doi.org/10.7554/eLife.31510.020

pore diameters. Below a critical pore diameter, the conductance is very low, whereas above it, it rises nearly linearly with diameter. Furthermore, the mutant shows a larger conductance than the native Nsp1. Gratifyingly, the experimental and theoretical data are even in good quantitative agreement (see inset *Figure 3B*). Note that this correspondence is remarkable, given the simplicity of the model that merely assumes a critical FG domain density. In order to generate a closed-form, continuous function for the conductance $G(d)$, we fit the conductivities in *Figure 3—figure supplement 2* with smooth sigmoidal functions, substitute these in *Equation 2*, and plot the results together with the experimental and numerical data points in *Figure 3C*. The figure clearly illustrates that both the non-linear increase at small pore diameters as well as the near-linear increase in conductance at large pore sizes are nicely captured by the theoretical conductance relation, in close agreement with the numerical and experimental data points. Some deviations remain in the crossover region, e.g., near 20–30 nm in the Nsp1-S mutant data.

It is of interest to put the conductance values that we report here for biomimetic NPCs in perspective. Early patch clamp studies of whole NPCs in vivo showed that NPCs are permeable to ions (*Bustamante et al., 1995*; *Tonini et al., 1999*), and all papers on NPCs since then have mentioned the good permeability of NPCs to ions and small molecules. However, the conductance of a single NPC is actually quite low, with values of only 0.3–2 nS, which is roughly two orders of magnitude lower than unhindered ionic transport (*Bustamante et al., 1995*; *Tonini et al., 1999*). It is noteworthy that our biomimetic NPC, with only 1 type of Nups, viz. Nsp1, has a conductance of ~4 nS for 35 nm pores, which is quite close to the in vivo value of 0.3–2 nS, certainly in view of the simplicity of our biomimetic NPC. Real NPCs consists of several types of Nups with a varying charge and FG content that all may affect the ion flux.

## Nsp1-pores are selective whereas Nsp1-S-pores are not selective for transport receptors

A critical question is to determine whether the NPC biomimetic pores are functional and selective regarding the transport of proteins. We compared the translocation of yeast NTR Kap95 (95 kDa) and a non-NTR tetrameric protein tCherry of similar size (104 kDa, see Materials and methods) through Nsp1-coated pores of size 48 ± 3 nm (as measured by TEM) (*Frey et al., 2006*; *Frey and Görlich, 2007*). First, we show the control experiment where we added either Kap95 or tCherry to the *cis* side of a bare (uncoated) pore. We observed clear translocation events as blockade peaks in the conductance (*Figure 4A*). Each downward spike is a single protein translocation event with a characteristic translocation time (τ) and conductance blockade (ΔG, the amplitude of the spikes in *Figure 4A and B*). We use a custom-made Matlab script to analyze our data as described elsewhere (*Plesa and Dekker, 2015*). In *Figure 4C–F*, each translocation event was represented as a dot in the scatter diagram, which shows the conductance blockade versus translocation time. A log-normal fit of the translocation times yields an average τ = 0.29 ± 0.16 ms and 0.19 ± 0.11 ms (mean ±standard deviation, for N = 3 pores), for Kap95 (100 nM) and tCherry

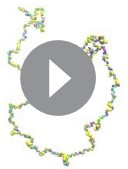

**Video 4.** A short trajectory of a coarse-grained one-bead-per-amino-acid molecular dynamics simulation of an isolated mutant Nsp1-S. The C-terminus is located at the top. The movie is prepared with the Visual Molecular Dynamics (VMD) software.
DOI: https://doi.org/10.7554/eLife.31510.021

(100 nM), respectively. The conductance blockade for Kap95 was 0.22 ± 0.07 nS and for tCherry 0.28 ± 0.11 nS. We thus conclude that, as expected from their similar size, the Kap95 and tCherry proteins translocate the bare nanopore with quite similar characteristics.

Next, we address the translocation though nanopores that were coated with Nsp1. Kap95 translocate through such pores with a most likely translocation time of τ = 5.2 ms±2.4 ms and an average conductance blockade of 0.31 ± 0.10 nS (N = 3). The conductance blockade is, as expected, of similar magnitude as for the bare pore. The most noteworthy difference is the significant increase in translocation time of Kap95 as it moves through the Nsp1-coated pore, indicating interactions between Nsp1 and Kap95. To probe whether these biomimetic NPCs also allow large non-NTR proteins to pass through, Kap95 was replaced by tCherry, a protein that is expected not to interact with the FG domains of NPCs. We found that passage of tCherry through the Nsp1-coated pores was essentially blocked: The tCherry translocation experiments yielded a significantly lower number of events (n = 90, compared to n = 917 for Kap95 in the same time window and at the same concentration; see *Figure 4E*). From these measurements, we conclude that the Nsp1-coated pore is selective: it does not allow tCherry to pass through efficiently, in contrast to the transport observed for Kap95.

One of the main objectives of this research was to address the importance of FG motifs in Nups such as Nsp1. Specifically, we ask ourselves whether the mutation of the hydrophobic FG motifs to the much less hydrophobic SG motifs affects the selective permeability barrier. To investigate this, we carried out translocation measurements with both Kap95 and tCherry on the Nsp1-S-coated nanopores. We successfully performed such experiments (*Figure 4D and F*), yielding an average translocation time for Kap95 of τ = 0.23 ± 0.13 ms (*Figure 4D*) and τ = 0.45 ± 0.23 ms for tCherry (*Figure 4F*), and a conductance blockade of 0.24 ± 0.10 nS and 0.32 ± 0.09 nS, respectively (N = 3). *Figure 4G and H* compare the translocation times for all cases. *Figure 4G* clearly shows the longer translocation time for Kap95 through Nsp1-coated pores, compared to both bare and Nsp1-S coated pores. *Figure 4H* shows that the tCherry translocation times are similar for bare and Nsp1-S-

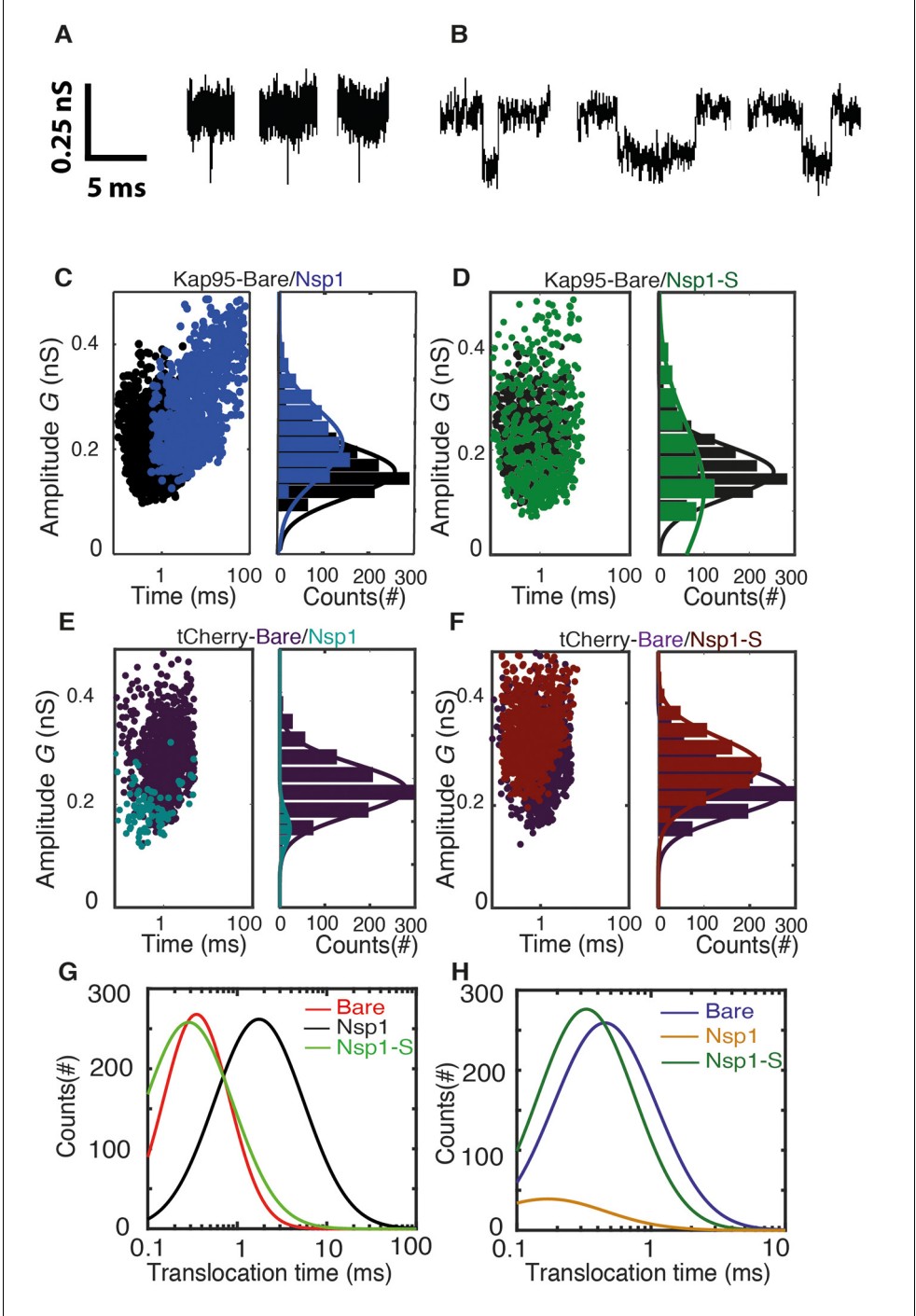

**Figure 4.** Transport and selectivity of the biomimetic NPCs. (**A**) Typical translocation event of Kap95 through a bare pore. Each spike signals a single kap95 that translocate the pore. (**B**) Examples of translocation events through the Nsp1-coated pores. Note that the events in panel B show translocation times of a few ms, in contrast to the sub-ms events of panel A. (**C**) Scatter plot comparing the translocation events of Kap95 through a bare pore (black; N = 1193) and through Nsp1-coated pores (blue; N = 917). The larger dwell time through the Nsp1-coated pore indicates a much slower transport than that through the bare pore. In C-F, the panels on the right show histograms of the conductance blockade. (**D**) Scatter diagram for Kap95 translocating through mutant Nsp1-S pores (green) (N = 505). The conductance blockade histograms show that the average conductance blockade level after Nsp1-S modification is comparable to that of bare pores. (**E**) Scatter plot comparing the translocation events of tCherry through a bare pore (Purple; N = 1000) and Nsp1 (Cyan; N = 90) coated pores. Note the low number of
*Figure 4 continued on next page*

*Figure 4 continued*

translocations of tCherry through Nsp1-coated pores. (**F**) Scatter plot comparing the translocation events of tCherry through a bare pore (Purple; N = 1000) and Nsp1-S (Red; N = 1000)-coated pores. (**G**) Translocation time distribution in lognormal format for translocation of Kap95 through bare, Nsp1-coated, and Nsp1-S-coated pores. (**H**) Same as (**G**) but for tCherry translocations through bare pore, Nsp1-, and Nsp1-S coated pores. Kap95 and tCherry concentrations were 100 nM.

DOI: https://doi.org/10.7554/eLife.31510.022

coated pores. We thus find that the Kap95 and tCherry actually translocate mutant-coated pores very well with short translocation times, similar to those for bare pores. The data show that the selectivity of the Nsp1-coated pores is lost when the hydrophobic FG-domains are replaced by the hydrophilic SG-domains.

The event frequency, that is, the number of translocation events per unit time, can be used as a figure of merit to quantify the selective behavior of Nsp1-coated and Nsp1-S-coated pores. *Figure 5A* compares the event frequency of Kap95 (blue, 100 nM) and tCherry (red, 100 nM) for translocations through 48 nm pores. The measured event rate of Kap95 was 1.7 ± 0.2 Hz for a bare pore, 1.4 ± 0.3 Hz for Nsp1-coated pores, and 1.4 ± 0.1 Hz for Nsp1-S-coated pores, i.e., Kap95 thus translocates through all pore types with a similar event rate. Additionally, we performed translocation measurements for varying Kap95 concentration (50–500 nM) through a Nsp1-coated pore, where we observed a constant baseline current at all Kap95 concentrations (*Figure 5—figure supplement 1*), contrasting to what would have been the case if large numbers of Kap95 would accumulate within the pore (*Lim et al., 2015*; *Kapinos et al., 2017*). Furthermore, we observed, as expected, a linear increase in the event frequency (*Figure 5—figure supplement 1*). Notably, in contrast to the finite 1.4 Hz event rate measured for Kap95, tCherry virtually fails to pass through the Nsp1-coated pores with an event frequency as low as 0.02 ± 0.04 Hz (at the same 100 nM concentration), while it translocates easily through the bare and Nsp1-S pores, with frequencies of 1.3 ± 0.2 Hz and 1.2 ± 0.3 Hz, respectively. Event frequencies for translocation through Nsp1-S-coated pores were also tested for various pore sizes in the range of 32 nm to 50 nm (*Figure 5—figure supplement 2*) for both Kap95 and tCherry. No clear diameter dependence of the selectivity was noted.

## Probing the NPC selectivity through MD simulations

To understand the mechanism behind the selectivity of the nanopores, we carried out coarse-grained MD simulations to calculate the energy barrier that the tCherry and Kap95 proteins have to overcome for transport through a 45nm-diameter pore that is lined with Nsp1 or with Nsp1-S. More specifically, we use the umbrella sampling method to calculate the potential of mean force (PMF) at every location along the central transport channel of the 45 nm nanopore (*Ghavami et al., 2016*). The PMF is the effective potential that the tCherry and Kap95 experiences due to the presence of the FG or SG domains, averaged over all conformations of the system. tCherry is simulated by using an inert (*Cardarelli et al., 2011*) sphere of radius 7.4 nm (see *Supplementary file 1*) while for Kap95 we use a sphere of radius 8.5 nm that is covered by 10 hydrophobic binding spots and a total charge of $-43e$ (*Kersey et al., 2012*) homogeneously distributed on the surface (*Tagliazucchi et al., 2013*).

*Figure 6* shows the potential of mean force (PMF) for tCherry and Kap95 particles at different *z*-positions along the central axis ($r = 0$) for the wildtype Nsp1 and mutant Nsp1-S pores. The energy barrier that the particles encounter can be seen as the work required for transport through the transport channel, and is defined as the difference between the maximum and minimum value of the PMF curve. In order to obtain the energy barrier, the PMF curves for positive and negative *z*-values are smoothened with a sixth-order polynomial function and these functions are used for further analysis. The PMF curves and the associated energy barriers can be understood in terms of the molecular interactions between the translocating particle and the FG-nups, which can be categorized into steric repulsion, hydrophobic/hydrophilic interactions, and electrostatic interactions (*Tagliazucchi et al., 2013*; *Ghavami et al., 2016*). Being an inert particle, tCherry only faces steric repulsion when entering the pore, whereas Kap95 is subjected to a higher steric repulsion (because it has a larger surface area (*Ghavami et al., 2016*)), but this is compensated by additional favorable hydrophobic and weak electrostatic interactions.

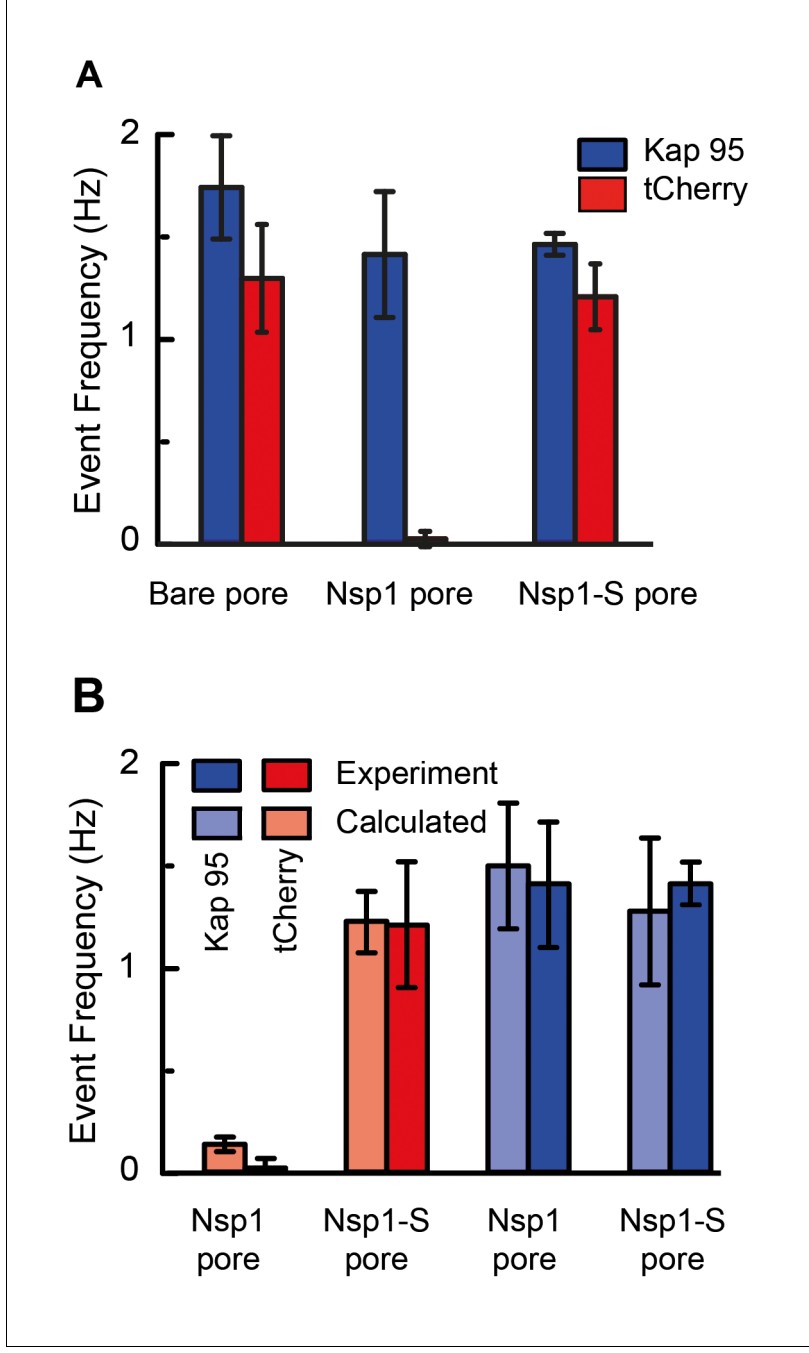

**Figure 5.** Selectivity for Nsp1, but not for Nps1-S biomimetic NPCs. (**A**) Event frequencies for Kap95 (blue) and tCherry (red) through bare pores, Nsp1-coated pores , and Nsp1-S-coated pores. The data show an NPC-like selectivity for Nsp1-coated pores where the passage of tCherry is inhibited whereas Kap95 can pass well through the pore, with an event frequency that is similar to the case of bare pore. Note that the Nsp1-S mutant pores allow both Kap95 and tCherry to pass through with a similar rate. The pore diameter was in all cases 48 ± 2 nm. Kap95 and tCherry concentrations were 100 nM. (**B**) A comparison between experimental event rate and calculated event rate using *Equation 4*. The error bar in the calculated event rate is computed based on the error in the energy barrier for the respective particle and pore combination.

DOI: https://doi.org/10.7554/eLife.31510.023

The following figure supplements are available for figure 5:

**Figure supplement 1.** Stable baseline conductance and increase of the event rate with Kap95 concentration for an Nsp1-coated pore.

*Figure 5 continued on next page*

*Figure 5 continued*

DOI: https://doi.org/10.7554/eLife.31510.024

**Figure supplement 2.** Event frequency versus diameter for Kap95 and tCherry through Nsp1-S coated nanopores.
DOI: https://doi.org/10.7554/eLife.31510.025

**Figure supplement 3.** Experimental event rate $\Gamma_0$ versus the computed energy barrier $\Delta E$, for tCherry and Kap95 in Nsp1 and Nsp1-S pores.
DOI: https://doi.org/10.7554/eLife.31510.026

If we focus on the wild-type Nsp1 pores, we see that the energy barrier for tCherry is high, 12 kJ mol$^{-1}$, i.e. almost 5 $k_B T$. This is entirely due to the strong steric hindrance that the inert tCherry particle experiences when it aims to pass through the high-density Nsp1 pore (100 mg/ml density, see *Figures 2C* and *3A*). In contrast, the PMF curve for Kap95 in the wild-type pore shows a drastically different behavior. Due to the hydrophobic binding sites and negative charge, the Kap95 particle is strongly attracted by the hydrophobic and weakly positively-charged Nsp1 meshwork, resulting in an energy well around |z| = 30 nm, which co-localizes with the high concentration of Nsp1 hydrophobic head groups (see *Figure 2—figure supplement 1* and *Supplementary file 2*). In order to complete translocation, the Kap95 has to overcome the energy barrier associated with the well, being equal to 6 kJ/mol$^{-1}$ (*Carpenter et al., 2014*), viz., a strong reduction compared to the steric Nsp1-barrier of tCherry of 12 kJ/mol$^{-1}$. Such an energy barrier is reminiscent of the entropic barrier reported by Rout et al (*Rout et al., 2003*).

In contrast to the big difference between tCherry and Kap95 in the Nsp1 energy landscape, there is almost no difference in the energy barriers for the two particles in Nsp1-S. Both curves show a

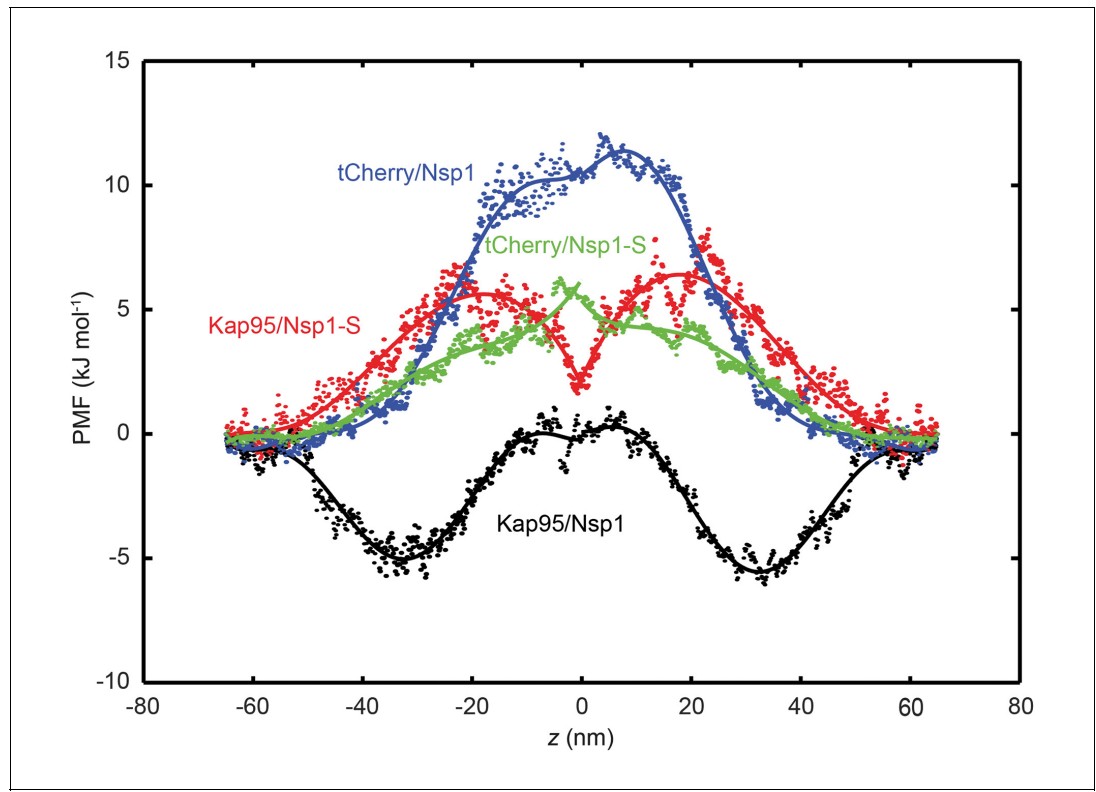

**Figure 6.** Potential of mean force (PMF) curves associated with transport of tCherry and Kap95 through Nsp1 and Nsp1-S coated pores. PMF values for tCherry and Kap95 particles at different positions along the central axis (*r* = 0). Kap95 in the Nsp1 and in Nsp1-S pore is represented in black and red, respectively, and tCherry in Nsp1 and Nsp1-S is shown in blue and green, respectively. Solid lines represent polynomial fits of 6$^{th}$ order for *z* < 0 and *z* > 0. The data show that the energy barrier that tCherry needs to overcome to move across the Nsp1-lined pore is approximately 12 kJ/mol, while for Kap95, it is much lower, 6 kJ/mol. For the Nsp1-S pore however, the barriers are very similar for Kap95 and tCherry, viz., 6.5 kJ/mol and 6.4 kJ/mol, respectively.

DOI: https://doi.org/10.7554/eLife.31510.027

rising energy profile when entering, with the Kap95 PMF rising stronger than tCherry for $|z| > 20$ nm, due to its bigger size. Around $z = 0$, Kap95 features a sharp drop in the potential of mean force, whereas the PMF for tCherry shows a small peak. This is associated with a higher (hydrophobic) protein density at $z = 0$, resulting in an increased steric repulsion for tCherry and an increased attraction for Kap95. Despite the fact that their specific energy profiles are different, the energy barriers calculated from the PMF curves are similar, 6.5 kJ mol$^{-1}$ and 6.4 kJ mol$^{-1}$ for tCherry and Kap95, respectively. Clearly, from an energetic point of view, the Nsp1-S pore thus is non-selective for the two particles.

In order to characterize the experimental translocation event frequency of tCherry and Kap95, we used an Arrhenius relation. The event frequency $\Gamma$ can be expressed as

$$\Gamma = \Gamma_0 \exp[-\Delta E / k_B T], \qquad (4)$$

in which $\Delta E$ is the energy barrier that the translocating particle has to overcome and $\Gamma_0$ is a proportionality constant that resembles the event frequency of the bare pore. The computed event rates are compared to the experimental event rates in *Figure 5B*, showing excellent agreement (for details see *Figure 5—figure supplement 3*): Kap95 translocates through the nanopore coated with Nsp1 at a rate of 1.5 ± 0.3 Hz, while tCherry features a much lower rate of 0.13 ± 0.03 Hz. In contrast, for the mutated pore, the frequencies are very similar: 1.3 ± 0.3 Hz for Kap95 and at 1.2 ± 0.1 Hz for tCherry. These results show that the permeability barrier of Nsp1-S is compromised, while Nsp1 allows Kap95 but not tCherry, featuring a clear transport selectivity.

## Discussion

The NPC central conduit, which controls all transport between nucleus and cytosol, is guarded by a barrier made up of intrinsically disordered Nups with FG-repeats (*Musser and Grünwald, 2016*). In this study, we examined the behavior of a minimalistic biomimetic NPC assembly with either Nsp1 or Nsp1-S domains tethered onto the inner surface of a solid-state nanopore. Our results provide the first experimental data for the ion conductance as a function of nanopore diameter (5 to 65 nm) for Nsp1 pores. Nsp1 pores are found to block the conductance stronger than Nsp1-S mutant pores. For small pore sizes the conductance is low (<4 nS), similar but slightly larger than natural NPCs (~1 nS) (*Bustamante et al., 1995*; *Tonini et al., 1999*). While these values are in fact remarkably close, the difference is not unexpected since we use only one type of FG-Nup domain, whereas the NPC consists of more than 10 different FG Nups with multiple copies. Beyond a non-linear transition regime at pore sizes of around 40 and 20 nm for Nsp1 and Nsp1-S, respectively, the conductance rises strongly, leading to a near-linear slope at larger pore sizes.

To shed light on these experimental findings, we carried out coarse-grained molecular dynamics simulations on nanopores of different sizes. As the model contains the exact amino-acid sequence of the FG domains, it captures the difference in cohesiveness between the mutant and wildtype Nsp1 and predicts their distribution inside the pore. Interestingly, the density in the Nsp1 pores exhibits a maximum at the pore center and remains significantly high throughout (see *Figure 2C* and *Figure 3A*), even for pore radii that are much larger than two times the Stokes radius of Nsp1 ($2R_S \sim 15$ nm). This can be partly attributed to the cohesive, sticky nature of the low-charge N-terminal 'head' segments of Nsp1 that are rich in hydrophobic FG-repeats and partly to the geometric confinement of the closely-spaced Nups in the channel (the grafting distance of 5.7 nm is considerably smaller than twice the Stokes radius ($2R_S \sim 15$ nm). This is consistent with the work of Zahn et al. (*Zahn et al., 2016*) who found Nsp1 brush heights of approximately 27 nm for similar grafting densities and with the work of Vovk et al., who found that for surface-tethered FG domains the brush height increases with decreasing grafting distance (*Vovk et al., 2016*). The Nsp1-S mutant lacks the large percentage of hydrophobic residues required for cohesive interactions (see *Video 4* and *Supplementary file 2*) leading to a remarkably uniform density distribution across the pore area (*Figure 3A*). The spatial structure of the FG and SG domains in the wildtype and mutant Nsp1 pores shows a striking difference with that of (non-glycosylated) Nup98 pores that exhibit a high-density, donut-like structure that are fully open at the center already at pore diameters of ~25 nm. This is due to the lower geometric confinement (the grafting distance was measured to be approximately equal to $2R_S = 8.2$ nm) and the much larger Nup98 cohesiveness. The latter reflects the smaller ratio of charged

over hydrophobic residues (0.2 for Nup98 versus 0.9 and 1.6 for Nsp1 and Nsp1-S, respectively), resulting in characteristic protein densities of 344, 74, and 52 mg/ml, respectively (see *Supplementary file 2*).

To link protein density to ionic conductance, we developed a phenomenological relation for the nanopore conductance *G(d)* which connects the in-silico time-averaged protein density distribution $\rho(r)$ to the effective ionic conductivity $\sigma$. By adopting a single critical protein concentration, the model successfully predicts the very low conductance at small pore sizes and the non-linear transition to an access-resistance-dominated conductance for larger pores. The conductance of the Nsp1-coated pores was found to be lower than those coated with Nsp1-S, which is a direct consequence of the sequence-dependent difference in spatial FG-domain distribution as discussed above. We also used the density-based conductance relation (*Equation 2*) to predict the conductance of the Nup98 nanopores published before (*Kowalczyk et al., 2011b*) (see *Figure 3—figure supplement 3*). Also in this case the experimental and computational data are in excellent agreement, demonstrating the broad applicability of the *G(d)* relation for Nup-nanopores that feature profoundly different protein density distributions.

The differences between the Nsp1 and Nsp1-S biomimetic pores testify to the importance of the hydrophobic residues for barrier-formation by FG domains. The formation of hydrogels (*Ribbeck and Görlich, 2001*; *Frey et al., 2006*) and surface-grafted FG domains (*Eisele et al., 2013*) similarly showed evidence of the FG motifs' role in establishing the cohesiveness of Nsp1 relative to Nsp1-S. Hexanediols have been shown to disrupt the permeability barrier of either human or yeast NPCs (*Lim et al., 2007*; *Patel et al., 2007*; *Jovanovic-Talisman et al., 2009*). Moreover, the FG motifs not only establish a cohesive structure (*Frey et al., 2006*; *Patel et al., 2007*; *Yamada et al., 2010*; *Hülsmann et al., 2012*), they also assist transport of NTRs through the NPC channel (*Rout and Wente, 1994*; *Bayliss et al., 1999*; *Bayliss et al., 2002*; *Isgro and Schulten, 2005*; *Port et al., 2015*).

To explore the role of Nsp1 cohesiveness in assisting transport, we tested our Nsp1 biomimetic NPCs for the selective permeability for Kap95 and tCherry. We observed that Kap95 translocates well through Nsp1-coated pores, with sizeable (few ms) dwell times, compared to the fast translocation through bare and Nsp1-S-coated pores. Note that the dwell time of ~5 ms is close to the 3 to 10 ms NTR passage times observed through the NPC (*Yang et al., 2004*; *Kubitscheck et al., 2005*), which is remarkable given the simplicity of our biomimetic NPCs. Contrary to Kap95, tCherry did hardly translocate through Nsp1-coated pores, indicating a clear selectivity of these biomimetic NPCs. Kap95 transport through Nsp1 was tested before as a model reporter for selective permeability in a variety of studies (*Frey and Görlich, 2007*; *Jovanovic-Talisman et al., 2009*; *Eisele et al., 2013*). In a previous report, Nsp1-based artificial NPCs were tested for selective behavior of various NTRs and NTR-cargo complexes which translocated effortlessly whereas the mutant version of NTR (with reduced binding affinity to the FG repeats) transported at a much lower rate (*Jovanovic-Talisman et al., 2009*). We observed a similar reduction in transport rate for tCherry compared to Kap95 translocations through the Nsp1-coated pores.

The molecular dynamics simulations provide clear mechanistic insight into the permeability and selectivity of the Nsp1 pores. The tCherry is an inert (i.e. non-interacting with FG repeats) particle, subject to steric repulsion from the FG domains and unable to counteract the barrier-forming hydrophobic interactions between the FG domains. Since the protein density in the Nsp1-pores (~100 mg/ml) is significantly higher than that in the Nsp1-S pore (~50 mg/ml), the energy barrier is almost two times as high (~12 kJ mol$^{-1}$ versus ~6.5 kJ mol$^{-1}$), resulting in Arrhenius-converted event rates that differ by an order of magnitude (0.13 and 1.22 Hz, respectively). In contrast to tCherry, Kap95 has hydrophobic binding sites and features a strong attraction to the hydrophobic residues of the Nsp1 FG domains. There is also a weak electrostatic attraction between the strongly negative Kap95 and the weakly positive FG domain. This together lowers the energy barrier from ~12 kJ mol$^{-1}$ for tCherry to ~6 kJ mol$^{-1}$ for Kap95 in the Nsp1 pore, associated with an increase in event rate from 0.13 to 1.50 Hz. We thus find that the Nsp1 pores are clearly selective, repelling inert particles but allowing transport receptors to pass through. The Nsp1-S pore, on the other hand, does not feature such a selective permeability barrier, allowing both tCherry as well as Kap95 to pass through, with event rates of 1.2 and 1.3 Hz, respectively. Clearly, when hydrophobic residues are replaced by hydrophilic residues, the permeability barrier is compromised and the selectivity vanishes.

To conclude, we have successfully built and modeled minimal NPCs based on solid-state nanopores with yeast Nsp1 and mutant Nsp1-S domains. We demonstrated a clear difference in the

conductance characteristics conferred by either Nsp1 or Nsp1-S. Translocation time and event rate analyses showed that Nsp1 is selective for the yeast importer Kap95 over tCherry, while Nsp1-S-coated pores lack this selective barrier, verifying that cohesive inter FG repeat interactions are required for transport selectivity. Major new biophysics insights into the underlying structural cause of all these experimental observations were obtained from coarse-grained molecular dynamics simulations of the FG Nup density distributions. It was shown that Nsp1 forms a high-density protein distribution with a pronounced maximum at the pore center, in contrast to a uniform and significantly less-dense protein distribution for Nsp1-S. The computed density-dependent conductance and translocation times of Kap95 and tCherry for the Nsp1 and Nsp1-S nanopores were found to be in excellent agreement with the experimental results. Our results identify a sequence-dependent spatial structure of the disordered FG-Nups that affects the conductance and highlights its key role in establishing the NPC's selective permeability.

## Materials and methods

### Solid-state nanopores

Solid-state nanopores were fabricated on free standing Silicon Nitride (SiN) membrane deposited on Silicon wafer as mentioned elsewhere in detail (*Janssen et al., 2012*). In brief, nanopore chips are built-up on a silicon wafer (100) with supporting deposited layers of Silicon dioxide and low-stress SiN. By employing UV-lithography, chemical etching and reactive-ion etching, the layers were etched away to end up with ~10 µm window of freestanding silicon nitride of 20 nm thickness. In this layer, a nanopore was drilled by electron beam using Transmission Electron Microscopy (TEM) operated at 300 kV. The focused electron beam was used to control the diameter of pore with a nanometer precision. After drilling with TEM, the pores were stored in a solution containing 50% (v/v) ethanol in Milli-Q water until usage. In our current work, we used pore diameter from 5 to 65 nm. Prior to measurement, each pores were painted with a layer of polydimethylsiloxane (PDMS) and baked for 2 hr at 70℃. PDMS layer reduce capacitive noise and offer better signal-to-noise properties (*Tabard-Cossa et al., 2007*). Nanopore chips were mounted on a custom made poly(methyl methacrylate) (PMMA) flow cell, after which the flow cell was filled with 150 mM KCl, 10 mM Tris- EDTA (1 mM) buffer at pH 7.6. The current was recorded with a electrophysiology patch clamp setup Axopatch 200B amplifier with a digitizer Digidata 1322A DAQ. We probe the transport electrically by monitoring the translocation of single proteins through 50 nm pores with a conductance of $G = 10$–$16$ nS. The concentration of Kap95 and tCherry used in translocation experiments was 100 nM unless stated otherwise. Note that measurements on nanopores with a diameter below the threshold are difficult due to a low signal-to-noise ratio at physiological salt conditions. The data was analyzed in a custom Transalyzer package in Matlab (*Plesa and Dekker, 2015*).

### Chemical modification of solid-state nanopore

The surface chemistry used to attach Nsp1 and Nsp1-S to the nanopore surface is shown in *Figure 1—figure supplement 1*. The nanopore chip was rinsed with water and Ethanol and treated with oxygen plasma for 60 s. The process cleans the surface from organic contaminants and makes the surface hydrophilic. The membranes were then (step 1) incubated with a 1% solution of APTES (3-aminopropyl-triethoxysilane) (Sigma) in pure methanol for 1 hr, followed by rinsing for 15 min in pure methanol. The chip was blow-dried under $N_2$ and baked at 100℃ for 60 min in order to fix the silane monolayer (*Wanunu and Meller, 2007*). The exposed amines were cross-linked with sulfo-SMCC (sulphosuccinimidyl-4-(N-maleimidomethyl)-cyclohexane-1-carboxylate) (2 mg no-weight capsules (Pierce)). Sulfo-SMCC has an amine-reactive NHS-ester and a maleimide group. A capsule of Sulfo-SMCC was dissolved in 1.5 ml PBS at pH 7.4 and nanopores were incubated in sulfo-SMCC solution overnight. The nanopores were rinsed with PBS to wash-off free sulfo-SMCC. The Nsp1 WT and mutant were stored in 7.3M guanidinium HCl and buffer exchanged to PBS, pH 7.4 and both the Nsp1 and Nsp1-S-mutants were treated with 1 mM TCEP for 30 mins to reactivate the SH-groups. The nanopores with maleimide were incubated with 120 nM Nsp1 and Nsp1-S for 1 hr. The C-terminal cysteine covalently bonds with the maleimide group to form a self-assembled layer of Nsp1 or Nsp1-S-mutant on the nanopore surface. The proteins Nsp1, Nsp1-S, Kap95, and tCherry

were purified by the methods described previously; for further details the reader is referred to ref (*Frey et al., 2006*; *Frey and Görlich, 2007*).

## Grafting density estimates of Nsp1 and Nsp1-S

We estimated the grafting density of the FG-Nups on the surface, and its importance for our results, in different ways:

### Estimate of the surface grafting density of the FG-Nups based on conductance

The idea of this approach is that one can estimate the number of, say, Nsp1 proteins that coat the pore from the drop in the conductance upon coating the pore, using the, independently measured, conductance blockade that is caused by a single protein as a reference. To pursue such an estimate, we translocated individual Nsp1 proteins through a bare pore (49 nm diameter) to estimate the ion blockade caused by a single Nsp1. The average conductance blockade of Nsp1 was found to be $0.54 \pm 0.15$ nS (cf. *Figure 1—figure supplement 4A*) with corresponding translocation times in the range of 0.1–5 ms. The average conductance blockade can be used to estimate the number of Nsp1 proteins that are blocking the ion flow through a nanopore of e.g. 48 nm size where the conductance dropped from 70 nS to 12 nS. This yields an estimate of $107 \pm 32$ for the number of Nsp1 proteins for this 48 nm pore. Assuming that a cylindrical pore volume of $\pi * (24 \text{ nm})^2 * 20 \text{ nm}$ confined these proteins, this line of reasoning provides a grafting density of about 1 Nsp1 per $28 \pm 8$ nm$^2$ (107 Nsp1 proteins per pore surface area of $2\pi * 24 * \text{nm} * 20 \text{ nm}$), resulting in a grafting distance of $5.7 \pm 0.8$ nm (assuming a close-packed triangular lattice).

These numbers yield an estimated Nsp1 density of about 320 mg/ml (107 Nsp1 proteins, each with a molecular weight of 65.7 kDa, in the cylindrical pore volume of $\pi * (24 \text{ nm})^2 * 20 \text{ nm}$). It is important to realize that this number is only a rough estimate based on a simplified geometry. For example, our MD simulations show that the Nps1 proteins 'spill out' out of the cylindrical nanopores such that an additional layer of Nsp1 is present above and below the nanopores, thus lowering the protein density in the cylindrical pore. Furthermore, the estimate neglects any intrinsic heterogeneities such as differences between the N-terminal part and C-terminal part of Nsp1. More accurate estimates for the Nsp1 density are provided by the MD simulations, see elsewhere in the paper.

Similarly, in independent experiments, individual Nsp1-S proteins were translocated through a bare pore (49 nm) to estimate the conductance blockade by single Nsp1-S proteins. The average conductance blockade of Nsp1-S was found to be $0.34 \pm 0.09$ nS (cf. *Figure 1—figure supplement 5A*). From the average conductance blockade, we again estimate the number of Nsp1-S proteins that were blocking the ion flow through a 50 nm nanopore where the conductance dropped from 70.3 nS to 34.6 nS. This yields an estimate of $105 \pm 30$ for the number of Nsp1-S proteins for a 50 nm pore. Assuming that a cylindrical pore volume of $\pi * (25 * \text{nm})^2 * 20 \text{ nm}$ confined these proteins, yields a surface coverage of about 1 Nsp1-S per $30 \pm 8$ nm$^2$ and a grafting distance of $5.9 \pm 0.8$ nm (assuming a close-packed triangular lattice).

### Estimate of the surface grafting density of the FG-Nups based on QCM-D surface coating experiments

An independent estimate of the grafting density can be obtained using QCM-D, a method that measures the accumulated mass upon coating a surface. The binding of FG-Nups through its C-terminal cysteine group to maleimide (Suflo-SMCC) on the surface was monitored versus time using QCM-D (QSense – QE401, Biolin Scientific AB, Sweden). The QCM-D shift upon exposure of Nsp1 (1 µM) or Nsp1-S (1 µM) to the functionalized silicon nitride surface is shown in *Figure 1—figure supplement 6*. QCM-D measures the shift in resonance frequency $\Delta f$ and the dissipation $\Delta D$ of the crystal due to the increased mass. The frequency shift was recorded for more than 5000 s, and no change in the frequency was observed upon flushing PBS to wash away any unbound proteins (flow rates 20 µl/min). The frequency shifts, $\Delta f = -60 \pm 5$ Hz (N = 3) for Nsp1 and $\Delta f = -56 \pm 15$ Hz (N = 3) for Nsp1-S, are, within errors, equal to each other. Ignoring dissipation, we can use the Sauerbrey relation $\Delta m = - (C * p * \Delta f)$ [where p=3 is the crystal overtone, C = 17.7e-9 kg s/m$^2$ at f = 5 MHz is the Sauerbrey constant, and $\Delta m$ is the areal mass (kg/m$^2$) that is added due to the protein coverage] to estimate the surface grafting distance as $5.6 \pm 0.2$ nm for Nsp1 and

5.8 ± 0.9 nm for Nsp1-S. The finite dissipation indicates some viscoelastic behavior in the adsorbed layers, and hence the mass will not couple 100% to the oscillatory motion of the sensor. Accordingly, the mass will be somewhat underestimated by the above calculation, which would make the numbers for the grafting densities slightly larger. Overall, however, we conclude that the surface coverage of Nsp1 and Nsp1-S is very similar, as well as close to the estimates obtained from the conductance experiments.

### Estimate of the importance of the grafting density of the FG-Nups from MD simulations

The translocation analysis and the QCM-D measurements show that the grafting density for Nsp1 and Nsp1-S are almost the same. We adopted an average value of 5.7 nm for the grafting distance for the MD simulations (see the section on Coarse-grained molecular dynamics simulations below) for both Nsp1 and Nsp1-S pores. We performed a sensitivity analysis for Nsp1-S, to probe how sensitive our results depend on the grafting density. *Figure 1—figure supplement 7* depicts the density profiles of a 45 nm pore for grafting spacings 5.7 and 5.9 nm, showing that the density inside the pore only changes marginally. Using the density-based conductance relation (*Eqn. 2*) a conductance of 29.1 ± 2.7 nS and 30.6 ± 2.8 nS results for the grafting distances of 5.7 nm and 5.9 nm, respectively, showing that also the effect on the conductance data is negligible.

### Coating the nanopores in the presence of Guanidine hydrochloride (3M)

As a further confirmation that the grafting density of Nsp1 and Nsp1-S are similar, we coated the nanopores (~50 nm) with proteins in the presence of 3M Guanidine hydrochloride. Under these conditions both Nsp1 and Nsp1-S will be in the denatured state and one does not expect any FG-FG interaction to be relevant for Nsp1. The pores were washed with PBS and the ionic conductance with measurement buffer. The bare pore conductance dropped from 78 nS to 16 nS after coating with Nsp1, which is about 80% conductance blockade. Interestingly, this is a similar conductance blockade observed in our measurements obtained in the normal assembly protocol. For the case of Nsp1-S, the bare pore conductance dropped from 82 nS to 45 nS, i.e. a conductance blocked of about 45% - again similar. These measurements confirm that the grafting of Nsp1 and Nsp1-S in the presence of 3M Guanidine hydrochloride was not different compared to coating in PBS

### Dynamic light scattering (DLS) measurement of the hydrodynamic diameter

DLS measurements are performed in 70 μl standard cuvettes from Brand GMBH with a light path of 10 mm. Both Nsp1 (1 μM) and Nsp1-S (1 μM) were treated with 5 mM dithiothreitol (DTT) for 30 min at room temperature to reduce cystine and to prevent formation of dimers. 1 μM of Kap95 or tCherry, treated with 1 mM of TCEP in PBS at pH 7.4 for 30 min was used for DLS measurement. All the samples were measured in buffer containing 150 mM KCl, 10 mM Tris, and 1 mM EDTA at pH 7.6. For the measurements, a Zetasizer Nano ZS (Malvern) was used at room temperature. DLS recordings (*Supplementary file 1*) were repeated at least three times for each protein and the result reported is the average over these recordings where errors represent the standard deviation. Dynamic light scattering experiments were also performed under denatured conditions (6M Guanidinium hydrochloride) for both Nsp1 and Nsp1-S yielding hydrodynamic diameters of 19.2 ± 5.3 nm and 19.4 ± 6.6 nm respectively.

### Coarse-grained molecular dynamics simulations

The one-bead-per-amino-acid MD model used here accounts for the exact amino-acid sequence of the FG-nups, with each bead centered at the $C_\alpha$ positions of the polypeptide chain (*Ghavami et al., 2013*; *Ghavami et al., 2014*). The average mass of the beads is 120 Da. Each bond is represented by a stiff harmonic spring potential with a bond length of 0.38 nm. The bending and torsion potentials for this model were extracted from the Ramachandran data of the coiled regions of protein structures (*Ghavami et al., 2013*). Solvent molecules are not accounted for explicitly; we account for the solvent polarity and screening effect of ions through a modified Coulomb law. Solvent polarity is incorporated through a distance-dependent dielectric constant, and ionic screening is accounted for through Debye screening with a screening constant $k = 1.27$ nm$^{-1}$ corresponding to a 150 mM KCl solution. The hydrophobic interactions among the amino-acids are incorporated through a modified

Lennard-Jones potential accounting for hydrophobicity scales of all 20 amino-acids through normalized experimental partition energy data renormalized in a range of 0 to 1. For details of the method and its parametrization, the reader is referred to ref. (*Ghavami et al., 2014*)

MD simulations for the nanopores with diameters ranging from 22 to 60 nm were carried out using GROMACS 4.5.1. First, the systems were energy minimized to remove any overlap of the amino acid beads. Then all long-range forces were gradually switched on. The simulations were carried out for over $5 \times 10^7$ steps (with the first $5 \times 10^6$ steps ignored for extracting the end-result data), which was found to be long enough to have converged results in the density distribution inside the pores.

The time-averaged density calculations presented in the main text and supporting information (see *Figure 2—figure supplement 1* and *Figure 2—figure supplement 2*) were carried out by centering the nanopore in a 100 nm x 100 nm x 140 nm box, which was divided into discrete cells of volume $(0.5 \text{ nm})^3$ and the number density in each cell was recorded as a function of simulation time. Finally, the number density was averaged over the simulation time and multiplied with the mass of each bead to get the time-averaged 3D density profile. The 3D density was averaged in the circumferential direction to obtain two-dimensional (2D) $(r, z)$ density plots (as shown in *Figure 2C*). Finally, the radial density distribution $\rho(r)$ was obtained by averaging these 2D density maps in the vertical direction (as shown in *Figure 3A*).

To estimate the energy barrier for translocation of transport factors through the biomimetic NPC, we calculate potential–of-mean-force (PMF) curves along the transport channel of the nanopores using the umbrella sampling method (*Ghavami et al., 2016*). In the current study, the cargo molecule is fixed at regularly spaced positions along the reaction coordinate (the central axis) of the pore by means of a harmonic spring. The spacing between the axial positions is chosen to be 1.3 nm and the spring constant is set to 10 kJ/nm$^2$/mole. For each position, two different starting configurations of the proteins are used to obtain statistically meaningful results. CG molecular dynamics simulations were performed for each particle position and the spring extension was recorded. The obtained distance histograms were used to calculate the PMF curves through a weighted histogram analysis. For further details on the umbrella sampling method, the reader is referred to ref. (*Ghavami et al., 2016*)

## Amino acid sequences of Nsp1 and Nsp1-S

The exact amino-acid sequence of Nsp1 and Nsp1-S was as follows:

| Nsp1 | MSKHHHHSGHHHTGHHHHSGSHHHTGENLYFQGSNFNTPQQNKTPFSFGTA NNNSNTTNQNSSTGAGAFGTGQSTFGFNNSAPNNTNNANSSITPAFGSNNTGN TAFGNSNPTSNVFGSNNSTTNTFGSNSAGTSLFGSSSAQQTKSNGTAGGNTFGS SSLFNNSTNSNTTKPAFGGLNFGGGNNTTPSSTGNANTSNNLFGATANANKPAF SFGATTNDDKKTEPDKPAFSFNSSVGNKTDAQAPTTGFSFGSQLGGNKTVNEAA KPSLSFGSGSAGANPAGASQPEPTTNEPAKPALSFGTATSDNKTTNTTPSFSFGA KSDENKAGATSKPAFSFGAKPEEKKDDNSSKPAFSFGAKSNEDKQDGTAKPAFS FGAKPAEKNNNETSKPAFSFGAKSDEKKDGDASKPAFSFGAKPDENKASAT |
| --- | --- |
| | SKPAFSFGAKPEEKKDDNSSKPAFSFGAKSNEDKQDGTAKPAFSFGAKPAEKN NNETSKPAFSFGAKSDEKKDGDASKPAFSFGAKSDEKKDSDSSKPAFSFGTKS NEKKDSGSSKPAFSFGAKPDEKKNDEVSKPAFSFGAKANEKKESDESKSAFSF GSKPTGKEEGDGAKAAISFGAKPEEQKSSDTSKPAFTFGAQKDNEKKTETSC |
| Nsp1-S | MSKHHHHSGHHHTGHHHHSGSHHHTGENLYFQGSNSNTPQQNKTPSSSGT ANNNSNTTNQNSSTGAGASGTGQSTSGSNNSAPNNTNNANSSSTPASGSNN TGNTASGNSNPTSNSSGSNNSTTNTSGSNSAGTSSSGSSSAQQTKSNGTAGGN TSGSSSSSNNSTNSNTTKPASGGSNSGGGNNTTPSSTGNANTSNNSSGATAN ANKPASSSGATTNDDKKTEPDKPASSSNSSSGNKTDAQAPTTGSSSGSQSGGN KTSNEAAKPSSSSGSGSAGANPAGASQPEPTTNEPAKPASSSGTATSDNKTTNT TPSSSSGAKSDENKAGATSKPASSSGAKPEEKKDDNSSKPASSSGAKSNEDKQ DGTAKPASSSGAKPAEKNNNETSKPASSSGAKSDEKKDGDASKPASSSGAKPDEN KASATSKPASSSGAKPEEKKDDNSSKPASSSGAKSNEDKQDGTAKPASSSGAKPA EKNNNETSKPASSSGAKSDEKKDGDASKPASSSGAKSDEKKDSDSSKPASSSGTK SNEKKDSGSSKPASSSGAKPDEKKNDESSKPASSSGAKANEKKESDESKSASSSG SKPTGKEEGDGAKAASSSGAKPEEQKSSDTSKPASTSGAQKDNEKKTE<u>S</u>TSC |

Please note that there is an extra S at position 635 (underlined above) in Nsp1-S that is there due to an artefact introduced while designing the gene synthesis used for cloning the SG mutant. While

nonideal, we are confident that – in comparison to the drastic changes made by mutating basically all hydrophobic sites to serines – the contribution of this one extra serine is negligible.

## Information on the tCherry control protein

A tCherry protein was chosen as a good reference to Kap95 because of its similar size and because it is a hydrophilic cytosolic protein that is expected to be inert regarding the interactions with hydrophobic Nups. In our study, tCherry, a tetramer of mCherry, was created by restoring the original tetramerization interface of DsRed (a tetrameric parent of mCherry) into mCherry (which originally was intended to be a monomeric derivative of DsRed). This operation results in a constitutively tetrameric protein with fluorescent properties similar to mCherry. According to our data, the protein is very hydrophilic, inert and – in contrast to DsRed itself – very well behaved.

The monomer sequence is: GTGMASSEDIIKEFMRFKVRMEGSVNGHEFEIEGEGEGRPYEGTQTAK LKVTKGGPLPFAWDILSPQFMYGSKAYVKHPADIPDYLKLSFPEGFKWERVMNFEDGGVVTVTQD SSLQDGEFIYKVKLIGVNFPSDGPVMQKKTMGWEASTERMYPRDGVLKGEIHQALKLKDGGHYLAEVKTI YMAKKPVQLPGYYYVDIKLDITSHNEDYTIVEQYERAEGRHHLFL

## Acknowledgements

We would like to acknowledge funding support from the Zernike Institute for Advanced Materials (University of Groningen) and the UMCG (to AM), NanoNextNL (program 07A.05), FOM and the Netherlands Organization for Scientific Research (NWO/OCW) [part of the Frontiers of Nanoscience program,], and the ERC Advanced Grant SynDiv [grant number 669598] (to CD). ANA thanks Yaron Caspi and Jakub Wiktor for valuable discussions and comments on the manuscript.

## Additional information

### Funding

| Funder | Grant reference number | Author |
| --- | --- | --- |
| Zernike Institute for Advanced Materials, University of Groningen | | Ankur Mishra |
| University Medical Center Groningen | | Ankur Mishra |
| NanoNextNL | program 637 07A.05 | Cees Dekker |
| FOM and Netherlands Organization for Scientific Research | | Cees Dekker |
| ERC Advanced Grant SynDiv | grant number 669598 | Cees Dekker |

The funders had no role in study design, data collection and interpretation, or the decision to submit the work for publication.

### Author contributions

Adithya N Ananth, Ankur Mishra, Conceptualization, Formal analysis, Investigation, Writing—original draft, Writing—review and editing; Steffen Frey, Resources, Writing—review and editing; Arvind Dwarkasing, Roderick Versloot, Formal analysis, Investigation, Writing—review and editing; Erik van der Giessen, Supervision, Writing—review and editing; Dirk Görlich, Writing—original draft, Writing—review and editing; Patrick Onck, Cees Dekker, Conceptualization, Supervision, Funding acquisition, Writing—original draft, Writing—review and editing

### Author ORCIDs

Erik van der Giessen (iD) http://orcid.org/0000-0002-8369-2254
Dirk Görlich (iD) https://orcid.org/0000-0002-4343-5210
Cees Dekker (iD) http://orcid.org/0000-0001-6273-071X

Decision letter and Author response
Decision letter https://doi.org/10.7554/eLife.31510.032
Author response https://doi.org/10.7554/eLife.31510.033

## Additional files

### Supplementary files

• Supplementary file 1. Measured hydrodynamic diameters (nm) of proteins from dynamic light scattering (DLS) experiment.
DOI: https://doi.org/10.7554/eLife.31510.028

• Supplementary file 2. Molecular characteristics of Nsp1 (see *Video 3*), Nsp1-S (see *Video 4*) and Nup98 and the computed Stokes radii (in isolation).
DOI: https://doi.org/10.7554/eLife.31510.029

• Transparent reporting form
DOI: https://doi.org/10.7554/eLife.31510.030

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
