## [Decision Letter]

Thank you for submitting your article "Spatial structure of disordered proteins dictates conductance and selectivity in Nuclear Pore Complex mimics" for consideration by *eLife*. Your article has been favorably evaluated by Richard Aldrich (Senior Editor) and three reviewers, one of whom, Richard M Berry (Reviewer #1), is a member of our Board of Reviewing Editors. The following individual involved in review of your submission has agreed to reveal their identity: Jan T Liphardt (Reviewer #3).

The reviewers have discussed the reviews with one another and the Reviewing Editor has drafted this decision to help you prepare a revised submission.

Summary:

The authors directly probe the transport of ions, and indirectly the transport of inert molecules and transport receptors through a pore in a membrane designed to mimic key aspects of the nuclear pore complex. Three types of pore-in-membrane configurations were explored: (i) un-derivatized (i.e., bare) (ii) derivatized with the WT FG domain of the nucleoporin Nsp1 (FG-nanopores), and (iii) derivatized with this same FG domain of Nsp1, wherein the hydrophobic residues F,I, L and V were replaced by S (SG-nanopores). The authors deduced that "FG-nanopores showed a clear selectivity as transport receptors can translocate across the pore whereas other proteins cannot. SG mutant pores lack such selectivity." They then present coarse-grained molecular dynamics simulations that they assert "reveal that FG-pores exhibit a high-density, nonuniform protein distribution, in contrast to a uniform and significantly less-dense protein distribution in the SG-mutant." They finally "conclude that the sequence-dependent density distribution of disordered proteins inside the NPC plays a key role for its conductivity and selective permeability."

Essential revisions:

1) Better evidence is required than is currently presented in the paper that the grafting densities are similar for Nsp1-FG versus Nsp1-SG. Without this assurance, the control represented by Nsp1-SG is not useful. The grafting experiments in guanidinium HCl do not convincingly demonstrate that the grafting densities of the Nsp1-FG and Nsp1-SG entities were similar.

Ideally a direct measure of these densities is needed since their physical properties, even in the presence of guanidium HCl, may be quite distinct and may yield different grafting densities. If this is not possible, further discussion and justification is required.

1b) The closest the manuscript comes to this is the coating density estimate in the pore, estimated from conductance blockade. However this is badly explained and therefore does not inspire confidence. Is it simply given by the simulated fit to conductivity data or are there independent measures? Subsection “Estimate of the surface grafting density of the FG-Nups based on conductance” states: "The average conductance blockade can be used to estimate the number of Nsp1 proteins that are blocking the ion flow through a nanopore of e.g. 48 nm where the conductance dropped from 70 nS to 12 nS. This yields an estimate of 107 for the number of Nsp1 proteins for this 48 nm pore." An explanation is needed of how this estimate is made. The implication seems to be that it is *not* via the MD simulations, which would make this a circular argument. It needs to be clear what model was used and what if any the free parameters were and how they were estimated. Or better still, provide an independent experimental measure.

2) The described transport experiments are performed with Kap95 and with tCherry separately rather than with them mixed together. Real transport works with both transport receptors and proteins that are not destined to be transported present at the same time. Indeed Lim and co-workers (PMID: 28864541) has again just shown how important Kap95 likely is for establishing the barrier function. This is an important factor not tested in the present mimics. Please state whether the authors attempted to make measurements on mixtures, and if not why not ? If so, can they differentiate Kap95 transit from tCherry transit in such mixtures by their very different transit times?

3) InFigure 5—figure supplement 2 and other measurements, state whether the multiple measurements were made on a single pore or on different pores, and report on the reproducibility of measurements made on separately fabricated pores with the same grafting material (e.g. FG domain of Nsp1)?

4) Please either present information on the transit event rate and behavior as a function of Kap95 and mCherry concentration in solution, or explain why there is none. Again, real transport in the NPC has been shown to involve the binding of many transport receptors to a single NPC at any one time, which makes this an important question.

---

## [Author Response]

Essential revisions:1) Better evidence is required than is currently presented in the paper that the grafting densities are similar for Nsp1-FG versus Nsp1-SG. Without this assurance, the control represented by Nsp1-SG is not useful. The grafting experiments in guanidinium HCl do not convincingly demonstrate that the grafting densities of the Nsp1-FG and Nsp1-SG entities were similar.Ideally a direct measure of these densities is needed since their physical properties, even in the presence of guanidium HCl, may be quite distinct and may yield different grafting densities. If this is not possible, further discussion and justification is required.

To thoroughly address this issue, we have now carried out significant new experiments. Gratifyingly, they reconfirm the notion that the grafting densities are the same for Nsp1-FG and Nsp1-SG.

Specifically:

1) We did experiments where we translocated Nsp1-SG through bare pores to measure the conductance blockade that each of these proteins cause (now added to subsection “Grafting density estimates of Nsp1 and Nsp1-S” and see Figure 1—figure supplement 5), and from that, we estimated the protein density of Nsp1-SG-coated pores. This yielded a grafting distance of 5.9 ± 0.8 nm. Importantly, this number is basically the same as the corresponding number for the grafting distance of 5.7 ± 0.8 nm for Nsp1-FG proteins.

2) We measured the surface density of Nsp1-FG and Nsp1-SG coatings using QCM-D, also added to subsection “Grafting density estimates of Nsp1 and Nsp1-S” and see Figure 1—figure supplement 6. These data show that the frequency shift upon surface coating translates to a grafting distance that is similar between Nsp1-FG and Nsp1-SG within 5% , which is remarkably close. Based on the frequency shifts, we estimate grafting distances of 5.6 ± 0.2 nm for Nsp1-FG and 5.8 ± 0.9 nm for Nsp1-SG.

Both these experiments, now added to the Materials and methods section, thus show that the grafting densities are very similar for Nsp1-FG and Nsp1-SG.

Furthermore, we performed additional MD simulations on the Nsp1-S nanopore to rule out any bias (also added to subsection “Grafting density estimates of Nsp1 and Nsp1-S” and Figure 1—figure supplement 7), showing that the measured small variations in grafting distance have a negligible effect on the density and conductance.

1b) The closest the manuscript comes to this is the coating density estimate in the pore, estimated from conductance blockade. However this is badly explained and therefore does not inspire confidence. Is it simply given by the simulated fit to conductivity data or are there independent measures? Subsection “Estimate of the surface grafting density of the FG-Nups based on conductance” states: "The average conductance blockade can be used to estimate the number of Nsp1 proteins that are blocking the ion flow through a nanopore of e.g. 48 nm where the conductance dropped from 70 nS to 12 nS. This yields an estimate of 107 for the number of Nsp1 proteins for this 48 nm pore." An explanation is needed of how this estimate is made.

We apologize if our explanation of this methodology was unclear. The idea is that one can estimate the number of Nsp1 proteins that coat the pore from the drop in the conductance upon coating the pore, using the (independently measured) conductance blockade that is caused by a single protein as a reference. We have rephrased the text and hope that it is clear now.

The implication seems to be that it is not via the MD simulations, which would make this a circular argument. It needs to be clear what model was used and what if any the free parameters were and how they were estimated. Or better still, provide an independent experimental measure.

This is correct, the grafting density was *not* obtained via the MD simulations. And indeed, we now provide independent experimental measures of it (see our response to point 1a).

2) The described transport experiments are performed with Kap95 and with tCherry separately rather than with them mixed together. Real transport works with both transport receptors and proteins that are not destined to be transported present at the same time. Indeed Lim and co-workers (PMID: 28864541) has again just shown how important Kap95 likely is for establishing the barrier function. This is an important factor not tested in the present mimics. Please state whether the authors attempted to make measurements on mixtures, and if not why not ? If so, can they differentiate Kap95 transit from tCherry transit in such mixtures by their very different transit times?

We appreciate the comment and we are well aware of the work from Lim and co-workers that emphasizes the importance of Kap95 in the NPC. However, the validity of it is still a point of ongoing discussion in the field. Various groups have reported conflicting evidence in favour and against a kap-centric model for nuclear pore transport. For instance, Eisele et al., 2010; Milles et al., 2015; Zahn et al., 2016 reported no changes observed to FG-Nup structures upon interactions with transport factors (TFs), whereas moderate changes to FG-Nups were reported at high TF concentrations by Wagner et al., 2015. Importantly, we note that Lim et al. reported the relevance of Kap95 in the NPC for high concentrations of ~5 μM, whereas all our measurements are carried out at very low concentrations of Kap95, typically only 100 nM.

Moreover, we can in fact address the important Kap95 question for our assay experimentally, as follows: If Kap95 would assemble in the nanopore, it would shift the conductance baseline when we would add Kap95 to the solution, as any additional Kap95 that would occupy the pore would partly block the ionic current. However, in experiments where we continuously monitored the nanopore conductance, we did not observe any change in the baseline conductance of Nsp1-coated pores upon adding Kap95 to the ionic buffer. This clearly indicates that Kap95 does not accumulate in our NPC mimics at a concentration of 100 nM. The only change that we observed in these experiments was that the rate of single-protein translocation events depended on Kap95 concentration in a (trivial) linear fashion, consistent with the constant baseline conductance. We have now added these new data and comment on it in the main text of the manuscript.

Finally, about measuring on Kap95/tCherry mixtures: It is not possible to do the measurements on mixtures that are suggested in this comment by the referees, due to the nature of our essay. The reason is that these two proteins feature a very similar size (which is why tCherry is such a nice control for Kap95) and ionic current measurements therefore produce very similar signals, making it impossible to distinguish individual Kap95 events from tCherry events in a mixture. Indeed, the measured conductance blockades for Kap95, 0.22 ± 0.07 nS, and that for tCherry, 0.28 ± 0.11 nS, overlap significantly.

3) In Figure 5—figure supplement 2 and other measurements, state whether the multiple measurements were made on a single pore or on different pores, and report on the reproducibility of measurements made on separately fabricated pores with the same grafting material (e.g. FG domain of Nsp1)?

We now added the missing information in the current Figure 5—figure supplement 2.

4) Please either present information on the transit event rate and behavior as a function of Kap95 and mCherry concentration in solution, or explain why there is none.

All the nanopore translocation data were recorded at Kap95 and tCherry concentrations of 100 nM. We have now added these concentrations to the legends of Figure 4 and 5. Furthermore, we now measured the event rate as a function of Kap95 concentration, which displayed a linear dependence in **Figure 5-figure supplement 1**.

Again, real transport in the NPC has been shown to involve the binding of many transport receptors to a single NPC at any one time, which makes this an important question.

On the relevance of a putative binding of many transport receptors to a single NPC, we refer to our response to point 2 above.